# Improved Guarantees for $k$-means++ and $k$-means++ Parallel

**Konstantin Makarychev**[*]    **Aravind Reddy**    **Liren Shan**
Department of Computer Science
Northwestern University
Evanston, IL, USA

## Abstract

In this paper, we study $k$-means++ and $k$-means∥, the two most popular algorithms for the classic $k$-means clustering problem. We provide novel analyses and show improved approximation and bi-criteria approximation guarantees for $k$-means++ and $k$-means∥. Our results give a better theoretical justification for why these algorithms perform extremely well in practice.

## 1 Introduction

$k$-means clustering is one of the most commonly encountered unsupervised learning problems. Given a set of $n$ data points in Euclidean space, our goal is to partition them into $k$ clusters (each characterized by a center), such that the sum of squares of distances of data points to their nearest centers is minimized. The most popular heuristic for solving this problem is Lloyd's algorithm (Lloyd, 1982), often referred to simply as "the $k$-means algorithm".

Lloyd's algorithm uses iterative improvements to find a locally optimal $k$-means clustering. The performance of Lloyd's algorithm crucially depends on the quality of the initial clustering, which is defined by the initial set of centers, called a *seed*. Arthur and Vassilvitskii (2007) and Ostrovsky, Rabani, Schulman, and Swamy (2006) developed an elegant randomized seeding algorithm, known as the $k$-means++ algorithm. It works by choosing the first center uniformly at random from the data set and then choosing the subsequent $k - 1$ centers by randomly sampling a single point in each round with the sampling probability of every point proportional to its current cost. That is, the probability of choosing any data point $x$ is proportional to the squared distance to its closest already chosen center. This squared distance is often denoted by $D^2(x)$. Arthur and Vassilvitskii (2007) proved that the expected cost of the initial clustering obtained by $k$-means++ is at most $8 (\ln k + 2)$ times the cost of the optimal clustering i.e., $k$-means++ gives an $8 (\ln k + 2)$-approximation for the $k$-means problem. They also provided a family of $k$-means instances for which the approximation factor of $k$-means++ is $2 \ln k$ and thus showed that their analysis of $k$-means++ is almost tight.

Due to its speed, simplicity, and good empirical performance, $k$-means++ is the most widely used algorithm for $k$-means clustering. It is employed by such machine learning libraries as Apache Spark MLlib, Google BigQuery, IBM SPSS, Intel DAAL, and Microsoft ML.NET. In addition to $k$-means++, these libraries implement a scalable variant of $k$-means++ called $k$-means∥ (read "$k$-means parallel") designed by Bahmani, Moseley, Vattani, Kumar, and Vassilvitskii (2012). Somewhat surprisingly, $k$-means∥ not only works better in parallel than $k$-means++ but also slightly outperforms $k$-means++ in practice in the single machine setting (see Bahmani et al. (2012) and Figure 1 below). However, theoretical guarantees for $k$-means∥ are substantially weaker than for $k$-means++.

The $k$-means∥ algorithm makes $T$ passes over the data set (usually $T = 5$). In every round, it independently draws approximately $\ell = \Theta(k)$ random centers according to the $D^2$ distribution. After each round it recomputes the distances to the closest chosen centers and updates $D^2(x)$ for all $x$ in

---

[*]Author order is alphabetical.

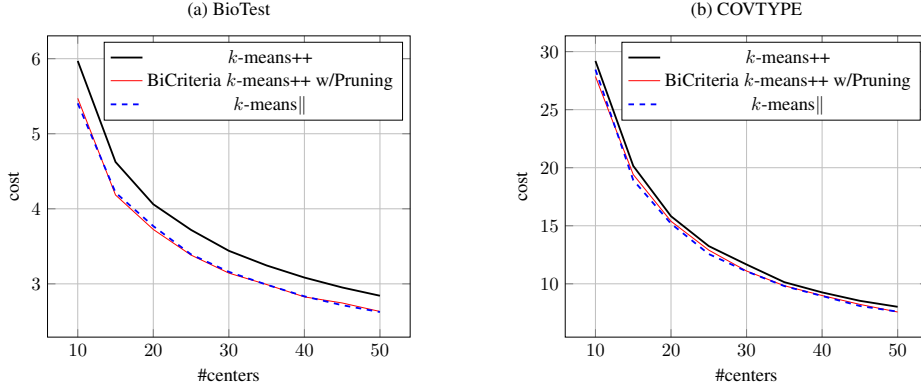

Figure 1: Performance of $k$-means++, $k$-means$\|$, and Bi-Criteria $k$-means++ with pruning on the BioTest and COVTYPE datasets. For $k = 10, 15, \cdots, 50$, we ran these algorithms for 50 iterations and took their average. We normalized the clustering costs by dividing them by $\text{cost}_{1000}(\mathbf{X})$.

the data set. Thus, after $T$ rounds, $k$-means$\|$ chooses approximately $T\ell$ centers. It then selects $k$ centers among $T\ell$ centers using $k$-means++.

**Our contributions.** In this paper, we improve the theoretical guarantees for $k$-means++, $k$-means$\|$, and Bi-Criteria $k$-means++ (which we define below).

First, we show that the expected cost of the solution output by $k$-means++ is at most $5(\ln k + 2)$ times the optimal solution's cost. This improves upon the bound of $8(\ln k + 2)$ shown by Arthur and Vassilvitskii (2007) and directly improves the approximation factors for several algorithms which use $k$-means++ as a subroutine like Local Search k-means++ (Lattanzi and Sohler, 2019). To obtain this result, we give a refined analysis of the expected cost of *covered clusters* (see Lemma 3.2 in Arthur and Vassilvitskii (2007) and Lemma 4.1 in this paper). We also show that our new bound on the expected cost of *covered clusters* is tight (see Lemma E.1).

Then, we address the question of why the observed performance of $k$-means$\|$ is better than the performance of $k$-means++. There are two possible explanations for this fact. (1) This may be the case because $k$-means$\|$ picks $k$ centers in two stages. At the first stage, it samples $\ell T \geq k$ centers. At the second stage, it prunes centers and chooses $k$ centers among $\ell T$ centers using $k$-means++. (2) This may also be the case because $k$-means$\|$ updates the distribution function $D^2(x)$ once in every round. That is, it recomputes $D^2(x)$ once for every $\ell$ chosen centers, while $k$-means++ recomputes $D^2(x)$ every time it chooses a center. In this paper, we empirically demonstrate that the first explanation is correct. First, we noticed that $k$-means$\|$ for $\ell \cdot T = k$ is almost identical with $k$-means++ (see Appendix A). Second, we compare $k$-means$\|$ with another algorithm which we call Bi-Criteria $k$-means++ with Pruning. This algorithm also works in two stages: At the Bi-Criteria $k$-means++ stage, it chooses $k + \Delta$ centers in the data set using $k$-means++. Then, at the Pruning stage, it picks $k$ centers among the $k + \Delta$ centers selected at the first stage again using $k$-means++. Our experiments on the standard data sets BioTest from KDD-Cup 2004 (Elber, 2004) and COVTYPE from the UCI ML repository (Dua & Graff, 2017) show that the performance of $k$-means$\|$ and Bi-Criteria $k$-means++ with Pruning are essentially identical (see Figures 1 and Appendix A).

These results lead to another interesting question: How good are $k$-means++ and $k$-means$\|$ algorithms that sample $k + \Delta$ instead of $k$ centers? The idea of oversampling using $k$-means++ was studied earlier in the literature under the name of *bi-criteria approximation*. Aggarwal, Deshpande, and Kannan (2009) showed that with constant probability, sampling $k+\Delta$ centers by $k$-means++ provides a constant-factor approximation if $\Delta \geq \beta k$ for some constant $\beta > 0$. Wei (2016) improved on this result by showing an expected approximation ratio of $8(1 + 1.618k/\Delta)$. Note that for bi-criteria algorithms we compare the expected cost of the clustering with $k + \Delta$ centers they produce and the cost of the optimal clustering with exactly $k$ centers.

In this paper, we show that the expected bi-criteria approximation ratio for $k$-means++ with $\Delta$ additional centers is at most the minimum of two bounds:

$$\text{(A) } 5\left(2 + \frac{1}{2e} + \ln\frac{2k}{\Delta}\right) \text{ for } 1 \leq \Delta \leq 2k; \text{ and (B) } 5\left(1 + \frac{k}{e\,(\Delta - 1)}\right) \text{ for } \Delta \geq 1$$

Both bounds are better than the bound by Wei (2016). The improvement is especially noticeable for small values of $\Delta$. More specifically, when the number of additional centers is $\Delta = k/\log k$, our approximation guarantee is $O(\log \log k)$ while Wei (2016) gives an $O(\log k)$ approximation.

We believe that our results for small values of $\Delta$ provide an additional explanation for why $k$-means++ works so well in practice. Consider a data scientist who wants to cluster a data set $\mathbf{X}$ with $k^*$ *true clusters* (i.e. $k^*$ latent groups). Since she does not know the actual value of $k^*$, she uses the *elbow method* (Boehmke & Greenwell, 2019) or some other heuristic to find $k$. Our results indicate that if she chooses slightly more number of clusters (for instance, $1.05k^*$), then she will get a constant bi-criteria approximation to the optimal clustering.

We also note that our bounds on the approximation factor smoothly transition from the regular ($\Delta = 0$) to bi-criteria ($\Delta > 0$) regime. We complement our analysis with an almost matching lower bound of $\Theta(\log(k/\Delta))$ on the approximation factor of $k$-means for $\Delta \leq k$ (see Appendix E).

We then analyze Bi-Criteria $k$-means$\|$ algorithm, the variant of $k$-means$\|$ that does not prune centers at the second stage. In their original paper, Bahmani, Moseley, Vattani, Kumar, and Vassilvitskii (2012) showed that the expected cost of the solution for $k$-means$\|$ with $T$ rounds and oversampling parameter $\ell$ is at most:

$$\frac{16}{1-\alpha}\mathrm{OPT}_k(\mathbf{X}) + \left(\frac{1+\alpha}{2}\right)^T \mathrm{OPT}_1(\mathbf{X}),$$

where $\alpha = \exp\left(-\left(1 - e^{-\ell/(2k)}\right)\right)$; $\mathrm{OPT}_k(\mathbf{X})$ is the cost of the optimal $k$-means clustering of $\mathbf{X}$; $\mathrm{OPT}_1(\mathbf{X})$ is the cost of the optimal clustering of $X$ with 1 center (see Section 2 for details). We note that $\mathrm{OPT}_1(\mathbf{X}) \gg \mathrm{OPT}_k(\mathbf{X})$. For $\ell = k$, this result gives a bound of $\approx 49\,\mathrm{OPT}_k(\mathbf{X}) + 0.83^T\mathrm{OPT}_1(\mathbf{X})$. Bachem, Lucic, and Krause (2017) improved the approximation guarantee for $\ell \geq k$ to

$$26\mathrm{OPT}_k(\mathbf{X}) + 2\left(\frac{k}{e\ell}\right)^T \mathrm{OPT}_1(\mathbf{X}).$$

In this work, we improve this bound for $\ell \geq k$ and also obtain a better bound for $\ell < k$. For $\ell \geq k$, we show that the cost of $k$-means$\|$ without pruning is at most

$$8\mathrm{OPT}_k(\mathbf{X}) + 2\left(\frac{k}{e\ell}\right)^T \mathrm{OPT}_1(\mathbf{X}).$$

For $\ell < k$, we give a bound of

$$\frac{5}{1 - e^{-\frac{\ell}{k}}}\,\mathrm{OPT}_k(\mathbf{X}) + 2\left(e^{-\frac{\ell}{k}}\right)^T \mathrm{OPT}_1(\mathbf{X})$$

Finally, we give a new parallel variant of the $k$-means++ algorithm, which we call *Exponential Race $k$-means++* ($k$-means++$_{\mathrm{ER}}$). This algorithm is similar to $k$-means$\|$. In each round, it also selects $\ell$ candidate centers in parallel (some of which may be dropped later) making one pass over the data set. However, after $T$ rounds, it returns exactly $k$ centers. The probability distribution of these centers is identical to the distribution of centers output by $k$-means++. The expected number of rounds is bounded as follows:

$$O\left(\frac{k}{\ell} + \log \frac{\mathrm{OPT}_1(\mathbf{X})}{\mathrm{OPT}_k(\mathbf{X})}\right).$$

This algorithm offers a unifying view on $k$-means++ and $k$-means$\|$. We describe it in Appendix F.

**Other related work.** Dasgupta (2008) and Aloise, Deshpande, Hansen, and Popat (2009) showed that $k$-means problem is NP-hard. Awasthi, Charikar, Krishnaswamy, and Sinop (2015) proved that it is also NP-hard to approximate $k$-means objective within a factor of $(1 + \varepsilon)$ for some constant $\varepsilon > 0$ (see also Lee, Schmidt, and Wright (2017)). We also mention that $k$-means was studied not only for Euclidean spaces but also for arbitrary metric spaces.

There are several known *constant* factor approximation algorithms for the $k$-means problem. Kanungo, Mount, Netanyahu, Piatko, Silverman, and Wu (2004) gave a $9 + \varepsilon$ approximation local search algorithm. Ahmadian, Norouzi-Fard, Svensson, and Ward (2019) proposed a primal-dual algorithm with an approximation factor of 6.357. This is the best known approximation for $k$-means. Makarychev, Makarychev, Sviridenko, and Ward (2016) gave constant-factor bi-criteria approximation algorithms based on linear programming and local search. Note that although these algorithms run in polynomial time, they do not scale well to massive data sets. Lattanzi and Sohler (2019)

provided a constant factor approximation by combining the local search idea with the $k$-means++ algorithm. Choo, Grunau, Portmann, and Rozhoň (2020) further improved upon this result by reducing the number of local search steps needed from $O(k \log \log k)$ to $O(k)$.

Independently and concurrently to our work, Rozhoň (2020) gave an interesting analysis for $k$-means$\|$ by viewing it as a *balls into bins* problem and showed that $O(\log n / \log \log n)$ rounds suffice to give a constant approximation with high probability.

## 2  Preliminaries

Given a set of points $\mathbf{X} = \{x_1, x_2, \cdots, x_n\} \subseteq \mathbb{R}^d$ and an integer $k \geq 1$, the $k$-means clustering problem is to find a set $C$ of $k$ centers in $\mathbb{R}^d$ to minimize

$$\text{cost}(\mathbf{X}, C) := \sum_{x \in \mathbf{X}} \min_{c \in C} \|x - c\|^2.$$

For any integer $i \geq 1$, let us define $\text{OPT}_i(\mathbf{X}) := \min_{|C|=i} \text{cost}(\mathbf{X}, C)$. Thus, $\text{OPT}_k(\mathbf{X})$ refers to the cost of the optimal solution for the $k$-means problem. Let $C^*$ denote a set of optimal centers. We use $\{P_i\}_{i=1}^k$ to denote the clusters induced by the center set $C^*$.

For any $\mathbf{Y} \subseteq \mathbf{X}$, the cost of $\mathbf{Y}$ with center set $C$, denoted by $\text{cost}(\mathbf{Y}, C) = \sum_{x \in \mathbf{Y}} \min_{c \in C} \|x - c\|^2$. The optimal cost for subset $\mathbf{Y}$ with $i$ centers is $\text{OPT}_i(\mathbf{Y})$. Let $\mu = \sum_{x \in \mathbf{Y}} x / |\mathbf{Y}|$ be the *centroid* of the cluster $\mathbf{Y}$. Then, we have the following closed form expression for the optimal cost of $\mathbf{Y}$ with one center (see Appendix B for proof),

$$\text{OPT}_1(\mathbf{Y}) = \sum_{x \in \mathbf{Y}} \|x - \mu\|^2 = \frac{\sum_{(x,y) \in \mathbf{Y} \times \mathbf{Y}} \|x - y\|^2}{2|\mathbf{Y}|}. \tag{1}$$

$k$**-means++ seeding:** The $k$-means++ algorithm samples the first center uniformly at random from the given points and then samples $k - 1$ centers sequentially from the given points with probability of each point being sampled proportional to its cost i.e. $\text{cost}(x, C)/\text{cost}(\mathbf{X}, C)$.

---

**Algorithm 1** $k$-means++ seeding

1: Sample a point $c$ uniformly at random from $\mathbf{X}$ and set $C_1 = \{c\}$.
2: **for** $t = 2$ **to** $k$ **do**
3:     Sample $x \in \mathbf{X}$ w.p. $\text{cost}(x, C_t)/\text{cost}(\mathbf{X}, C_t)$.
4:     $C_t = C_{t-1} \cup \{x\}$.
5: **end for**
6: **Return** $C_k$

---

$k$**-means$\|$ and** $k$**-means$\|_{\text{Pois}}$ seeding:** In the $k$-means$\|$ algorithm, the first center is chosen uniformly at random from $\mathbf{X}$. But after that, at each round, the algorithm samples each point independently with probability $\min\{\ell \cdot \text{cost}(x, C)/\text{cost}(\mathbf{X}, C), 1\}$ where $\ell$ is the *oversampling parameter* chosen by the user and it usually lies between $0.1k$ and $10k$. The algorithm runs for $T$ rounds (where $T$ is also a parameter chosen by the user) and samples around $\ell T$ points, which is usually strictly larger than $k$. This oversampled set is then weighted using the original data set $\mathbf{X}$ and a weighted version of $k$-means++ is run on this set to get the final $k$-centers. We only focus on the stage in which we get the oversampled set because the guarantees for the second stage come directly from $k$-means++.

For the sake of analysis, we also consider a different implementation of $k$-means$\|$, which we call $k$-means$\|_{\text{Pois}}$ (Algorithm 3). This algorithm differs from $k$-means$\|$ in that each point is sampled independently with probability $1 - \exp(-\ell \cdot \text{cost}(x, C)/\text{cost}(\mathbf{X}, C))$ rather than $\min\{\ell \cdot \text{cost}(x, C)/\text{cost}(\mathbf{X}, C), 1\}$. In practice, there is essentially no difference between $k$-means$\|$ and $k$-means$\|_{\text{Pois}}$, since $\ell \cdot \text{cost}(x, C)/\text{cost}(\mathbf{X}, C)$ is a very small number for all $x$ and thus the sampling probabilities for $k$-means$\|$ and $k$-means$\|_{\text{Pois}}$ are almost equal.

| **Algorithm 2** $k$-means$\|$ seeding | **Algorithm 3** $k$-means$\|_{\text{Pois}}$ seeding |
|---|---|
| 1: Sample a point $c$ uniformly from $\mathbf{X}$ and set $C_1 = \{c\}$ | 1: Sample a point $c$ uniformly from $\mathbf{X}$ and set $C_1 = \{c\}$ |
| 2: **for** $t = 1$ **to** $T$ **do** | 2: **for** $t = 1$ **to** $T$ **do** |
| 3:      Sample each point $x$ into $C'$ independently w.p. $\min\{1, \lambda_t(x)\}$ where $\lambda_t(x) = \ell \cdot \text{cost}(x, C_t)/\text{cost}(\mathbf{X}, C_t)$ | 3:      Sample each point $x$ into $C'$ independently w.p. $1 - e^{-\lambda_t(x)}$ where $\lambda_t(x) = \ell \cdot \text{cost}(x, C_t)/\text{cost}(\mathbf{X}, C_t)$ |
| 4:      Let $C_{t+1} = C_t \cup C'$. | 4:      Let $C_{t+1} = C_t \cup C'$. |
| 5: **end for** | 5: **end for** |

In the rest of the paper, we focus only on the *seeding* step of $k$-means++, $k$-means$\|$, and $k$-means$\|_{\text{Pois}}$ and ignore Lloyd's iterations as the approximation guarantees for these algorithms come entirely from the seeding step.

## 3   General framework

In this section, we describe a general framework we use to analyze $k$-means++ and $k$-means$\|_{\text{Pois}}$. Consider $k$-means++ or $k$-means$\|_{\text{Pois}}$ algorithm. Let $C_t$ be the set of centers chosen by this algorithm after step $t$. For the sake of analysis, we assume that $C_t$ is an ordered set or list of centers, and the order of centers in $C_t$ is the same as the order in which our algorithm chooses these centers. We explain how to order centers in $k$-means$\|_{\text{Pois}}$ algorithm in Section 6. We denote by $T$ the stopping time of the algorithm. Observe that after step $t$ of the algorithm, the probabilities of choosing a new center in $k$-means++ or a batch of new centers in $k$-means$\|_{\text{Pois}}$ are defined by the current costs of points in $\mathbf{X}$ which, in turn, are completely determined by the current set of centers $C_t$. Thus, the states of the algorithm form a Markov chain.

In our analysis, we fix the optimal clustering $\mathcal{P} = \{P_1, \ldots, P_k\}$ (if this clustering is not unique, we pick an arbitrary optimal clustering). The optimal cost of each cluster $P_i$ is $\text{OPT}_1(P_i)$ and the optimal cost of the entire clustering is $\text{OPT}_k(\mathbf{X}) = \sum_{i=1}^k \text{OPT}_1(P_i)$.

Following the notation in Arthur and Vassilvitskii (2007), we say that a cluster $P_i$ is *hit* or *covered* by a set of centers $C$ if $C \cap P_i \neq \varnothing$; otherwise, we say that $P_i$ is *not hit* or *uncovered*. We split the cost of each cluster $P_i$ into two components which we call the covered and uncovered costs of $P_i$. For a given set of centers $C$,

$$\text{The covered or hit cost of } P_i, \qquad H(P_i, C) := \begin{cases} \text{cost}(P_i, C), & \text{if } P_i \text{ is covered by } C \\ 0, & \text{otherwise.} \end{cases}$$

$$\text{The uncovered cost of } P_i, \qquad U(P_i, C) := \begin{cases} 0, & \text{if } P_i \text{ is covered by } C \\ \text{cost}(P_i, C), & \text{otherwise.} \end{cases}$$

Let $H(\mathbf{X}, C) = \sum_{i=1}^k H(P_i, C)$ and $U(\mathbf{X}, C) = \sum_{i=1}^k U(P_i, C)$. Then,

$$\text{cost}(\mathbf{X}, C) = H(\mathbf{X}, C) + U(\mathbf{X}, C).$$

For the sake of brevity, we define $\text{cost}_t(\mathbf{Y}) := \text{cost}(\mathbf{Y}, C_t)$ for any $\mathbf{Y} \subseteq \mathbf{X}$, $H_t(P_i) := H(P_i, C_t)$, and $U_t(P_i) := U(P_i, C_t)$. In Section 4, we show that for any $t$, we have $\mathbb{E}[H_t(\mathbf{X})] \leq 5\text{OPT}_k(\mathbf{X})$, which is an improvement over the bound of $8\text{OPT}_k(\mathbf{X})$ given by Arthur and Vassilvitskii (2007). Then, in Sections 5 and 6, we analyze the expected uncovered cost $U(\mathbf{X}, C_T)$ for $k$-means++ and $k$-means$\|$ algorithms.

Consider a center $c$ in $C$. We say that $c$ is a *miss* if another center $c'$ covers the same cluster in $\mathcal{P}$ as $c$, and $c'$ appears before $c$ in the ordered set $C$. We denote the number of misses in $C$ by $M(C)$ and the the number of clusters in $\mathcal{P}$ not covered by centers in $C$ by $K(C)$.

Observe that the stochastic processes $U_t(P_i)$ with discrete time $t$ are non-increasing since the algorithm never removes centers from the set $C_t$ and therefore the distance from any point $x$ to $C_t$ never increases. Similarly, the processes $H_t(P_i)$ are non-increasing after the step $t_i$ when $P_i$ is covered first time. In this paper, we sometimes use a proxy $\widetilde{H}_t(P_i)$ for $H_t(P_i)$, which we define as follows. If $P_i$ is covered by $C_t$, then $\widetilde{H}_t(P_i) = H_{t_i}(P_i)$, where $t_i \leq t$ is the first time when

$P_i$ is covered by $C_t$. If $P_i$ is not covered by $C_t$, then $\widetilde{H}_t(P_i) = 5\mathrm{OPT}_1(P_i)$. It is easy to see that $H_t(P_i) \leq \widetilde{H}_{t'}(P_i)$ for all $t \leq t'$. In Section 4, we also show that $\widetilde{H}_t(P_i)$ is a supermartingale i.e., $\mathbb{E}[\widetilde{H}_{t'}(P_i) \mid C_t] \leq \widetilde{H}_t(P_i)$ for all $t \leq t'$.

## 4 Bound on the cost of covered clusters

In this section, we improve the bound by Arthur and Vassilvitskii (2007) on the expected cost of a covered cluster in $k$-means++. Our bound also works for $k$-means$\|_{\mathrm{Pois}}$ algorithm. Pick an arbitrary cluster $P_i$ in the optimal solution $\mathcal{P} = \{P_1, \ldots, P_k\}$ and consider an arbitrary state $C_t = \{c_1, \ldots, c_N\}$ of the $k$-means++ or $k$-means$\|_{\mathrm{Pois}}$ algorithm. Let $D_{t+1}$ be the set of new centers the algorithm adds to $C_t$ at step $t$ (for $k$-means++, $D_{t+1}$ contains only one center). Suppose now that centers in $D_{t+1}$ cover $P_i$ i.e. $D_{t+1} \cap P_i \neq \varnothing$. We show that the expected cost of cluster $P_i$ after step $(t+1)$ conditioned on the event $\{D_{t+1} \cap P_i \neq \varnothing\}$ and the current state of the algorithm $C_t$ is upper bounded by $5\mathrm{OPT}_1(P_i)$ i.e.

$$\mathbb{E}\left[\mathrm{cost}(P_i, C_{t+1}) \mid C_t, \{D_{t+1} \cap P_i \neq \varnothing\}\right] \leq 5\mathrm{OPT}_1(P_i). \tag{2}$$

We now prove the main lemma.

**Lemma 4.1.** *Consider an arbitrary set of centers $C = \{c_1, \ldots, c_N\} \subseteq \mathbb{R}^d$ and an arbitrary set $P \subseteq \mathbf{X}$. Pick a random point $c$ in $P$ with probability $\Pr(c = x) = \mathrm{cost}(x, C)/\mathrm{cost}(P, C)$. Let $C' = C \cup \{c\}$. Then, $\mathbb{E}_c\left[\mathrm{cost}(P, C')\right] \leq 5\mathrm{OPT}_1(P)$.*

**Remarks:** Lemma 3.2 in the paper by Arthur and Vassilvitskii (2007) gives a bound of $8\mathrm{OPT}_1(P)$. We also show in Appendix E that our bound is tight (see Lemma E.1).

*Proof.* The cost of any point $y$ after picking center $c$ equals the squared distance from $y$ to the set of centers $C' = C \cup \{c\}$, which in turn equals $\min\{\mathrm{cost}(y, C), \|y - c\|^2\}$. Thus, if a point $x \in P$ is chosen as a center, then the cost of point $y$ equals $\min\{\mathrm{cost}(y, C), \|x - y\|^2\}$. Since $\Pr(c = x) = \mathrm{cost}(x, C)/\mathrm{cost}(P, C)$, we have

$$\mathbb{E}_c\left[\mathrm{cost}(P, C')\right] = \sum_{\substack{x \in P \\ y \in P}} \frac{\mathrm{cost}(x, C)}{\mathrm{cost}(P, C)} \cdot \min\{\mathrm{cost}(y, C), \|x - y\|^2\}.$$

We write the right hand side in a symmetric form with respect to $x$ and $y$. To this end, we define function $f$ as follows:

$$f(x, y) = \mathrm{cost}(x, C) \cdot \min\left\{\|x - y\|^2, \mathrm{cost}(y, C)\right\} + \mathrm{cost}(y, C) \cdot \min\left\{\|x - y\|^2, \mathrm{cost}(x, C)\right\}.$$

Note that $f(x, y) = f(y, x)$. Then,

$$\mathbb{E}_c\left[\mathrm{cost}(P, C')\right] = \frac{1}{2\mathrm{cost}(P, C)} \sum_{(x,y) \in P \times P} f(x, y).$$

We now give an upper bound on $f(x, y)$ and then use this bound to finish the proof of Lemma 4.1.

**Lemma 4.2.** *For any $x, y \in P$, we have $f(x, y) \leq 5\min\left\{\mathrm{cost}(x, C), \mathrm{cost}(y, C)\right\}\|x - y\|^2$.*

We defer the proof of this lemma to Appendix C. We use Lemma 4.2 to bound the expected cost of $P$. Let $\phi^*$ be a vector in $\mathbb{R}^P$ with $\phi_x^* = \mathrm{cost}(x, C)$ for any $x \in P$. Then, $f(x, y) \leq 5\min\left\{\phi_x^*, \phi_y^*\right\}\|x - y\|^2$. Since $\mathrm{cost}(P, C) = \sum_{z \in P} \phi_z^*$, we have

$$\mathbb{E}_c\left[\mathrm{cost}(P, C')\right] \leq \underbrace{\frac{5\sum_{(x,y) \in P \times P} \min\left\{\phi_x^*, \phi_y^*\right\}\|x - y\|^2}{2\sum_{z \in P} \phi_z^*}}_{5F(\phi^*)}.$$

For arbitrary vector $\phi \in \mathbb{R}^P_{\geq 0}$, define the following function:

$$F(\phi) = \frac{\sum_{(x,y) \in P \times P} \min\left\{\phi_x, \phi_y\right\}\|x - y\|^2}{2\sum_{z \in P} \phi_z}. \tag{3}$$

We have $\mathbb{E}_c\left[\text{cost}(P, C')\right] \leq 5F(\phi^*)$. Thus, to finish the proof of Lemma 4.1, it suffices to show that $F(\phi) \leq \text{OPT}_1(P)$ for every $\phi \geq 0$ and particularly for $\phi = \phi^*$. By Lemma C.1 (which we state and prove in Appendix), function $F(\phi)$ is maximized when $\phi \in \{0, 1\}^P$. Let $\phi^{**}$ be a maximizer of $F(\phi)$ in $\{0, 1\}^P$ and $P' = \{x \in P : \phi_x^{**} = 1\}$. Observe that

$$F(\phi^{**}) = \frac{\sum_{(x,y) \in P' \times P'} \|x - y\|^2}{2|P'|} = \text{OPT}_1(P').$$

Here we used the closed form expression (1) for the optimal cost of cluster $P'$. Since $P' \subset P$, we have $\text{OPT}_1(P') \leq \text{OPT}_1(P)$. Thus, $F(\phi^*) \leq F(\phi^{**}) \leq \text{OPT}_1(P)$. □

Replacing the bound in Lemma 3.2 from the analysis of Arthur & Vassilvitskii (2007) by our bound from Lemma 4.1 gives the following result (see also Lemma D.5).

**Theorem 4.3.** *The approximation factor of k-means++ is at most* $5(\ln k + 2)$.

We now state an important corollary of Lemma 4.1.

**Corollary 4.4.** *For every* $P \in \mathcal{P}$, *the process* $\widetilde{H}_t(P)$ *for k-means++ is a supermartingale i.e.,* $\mathbb{E}\left[\widetilde{H}_{t+1}(\mathbf{X}) \mid C_t\right] \leq \widetilde{H}_t(\mathbf{X})$.

*Proof.* The value of $\widetilde{H}_t(\mathbf{X})$ changes only if at step $t$, we cover a yet uncovered cluster $P$. In this case, the value of $\widetilde{H}_{t+1}(P)$ changes by the new cost of $P$ minus $5\text{OPT}(P)$. By Lemma 4.1 this quantity is non-positive in expectation. □

Since the process $\widetilde{H}_t(P)$ is a supermartingale, we have $\mathbb{E}[\widetilde{H}_t(P)] \leq \widetilde{H}_0(P) = 5\text{OPT}_1(P)$. Hence, $\mathbb{E}[H_t(P)] \leq \mathbb{E}[\widetilde{H}_t(P)] = 5\text{OPT}_1(P)$. Thus, $\mathbb{E}[H_t(X)] \leq 5\text{OPT}_k(\mathbf{X})$. Since $\text{cost}_t(\mathbf{X}) = H_t(\mathbf{X}) + U_t(\mathbf{X})$ and we have a bound on the expectation of the covered cost, $H_t(\mathbf{X})$, in the remaining sections, we shall only analyze the uncovered cost $U_t(\mathbf{X})$.

## 5 Proof overview for bi-criteria $k$-means++

In this section, we give a bi-criteria approximation guarantee for $k$-means++.

**Theorem 5.1.** *Let* $\text{cost}_{k+\Delta}(\mathbf{X})$ *be the cost of the clustering with* $k + \Delta$ *centers sampled by the k-means++ algorithm. Then, for* $\Delta \geq 1$, *the expected cost* $\mathbb{E}\left[\text{cost}_{k+\Delta}(\mathbf{X})\right]$ *is upper bounded by (below* $(a)^+$ *denotes* $\max(a, 0)$*).*

$$\min\left\{2 + \frac{1}{2e} + \left(\ln\frac{2k}{\Delta}\right)^+, 1 + \frac{k}{e(\Delta - 1)}\right\} 5\text{OPT}_k(\mathbf{X}).$$

Note that the above approximation guarantee is the minimum of two bounds: (1) $2 + \frac{1}{2e} + \ln\frac{2k}{\Delta}$ for $1 \leq \Delta \leq 2k$; and (2) $1 + \frac{k}{e(\Delta-1)}$ for $\Delta \geq 1$. The second bound is stronger than the first bound when $\Delta/k \gtrsim 0.085$. We prove this theorem in Appendix D. Here we present a high level overview of the proof. Our proof consists of three steps.

First, we prove bound (2) on the expected cost of the clustering returned by $k$-means++ after $k + \Delta$ rounds. We argue that the expected cost of the covered clusters is bounded by $5\text{OPT}_k(\mathbf{X})$ (see Section 3) and thus it is sufficient to bound the expected cost of uncovered clusters. Consider an optimal cluster $P \in \mathcal{P}$. We need to estimate the probability that it is not covered after $k + \Delta$ rounds. We upper bound this probability by the probability that the algorithm does not cover $P$ before it makes $\Delta$ misses (note: after $k + \Delta$ rounds $k$-means++ must make at least $\Delta$ misses).

In this overview, we make the following simplifying assumptions (which turn out to be satisfied in the worst case for bi-criteria $k$-means++): Suppose that the uncovered cost of cluster $P$ does not decrease before it is covered and equals $U(P)$ and, moreover, the total cost of all covered clusters almost does not change and equals $H(\mathbf{X})$ (this may be the case if one large cluster contributes most of the covered cost, and that cluster is covered at the first step of $k$-means++). Under these assumptions, the probability that $k$-means++ chooses $\Delta$ centers in the already covered clusters and does not choose

a single center in $P$ equals $(H(\mathbf{X})/(U(P) + H(\mathbf{X})))^\Delta$. If $k$-means++ does not choose a center in $P$, the *uncovered* cost of cluster $P$ is $U(P)$; otherwise, the *uncovered* cost of cluster $P$ is $0$. Thus, the expected *uncovered cost* of $P$ is $(H(\mathbf{X})/(U(P) + H(\mathbf{X})))^\Delta U(P)$. It is easy to show that $(H(\mathbf{X})/(U(P) + H(\mathbf{X})))^\Delta U(P) \leq H(\mathbf{X})/(e(\Delta - 1))$. Thus, the expected *uncovered cost* of all clusters is at most

$$\frac{k}{(e(\Delta - 1))} \mathbb{E}[H(\mathbf{X})] \leq \frac{k}{(e(\Delta - 1))} 5\mathrm{OPT}_k(\mathbf{X}).$$

Then, we use ideas from Arthur and Vassilvitskii (2007), Dasgupta (2013) to prove the following statement: Let us count the cost of uncovered clusters only when the number of misses after $k$ rounds of $k$-means++ is greater than $\Delta/2$. Then the expected cost of uncovered clusters is at most $O(\log(k/\Delta)) \cdot \mathrm{OPT}_k(\mathbf{X})$. That is, $\mathbb{E}[H(U_k(\mathbf{X}) \cdot \mathbf{1}\{M(C_k) \geq \Delta/2\}] \leq O(\log(k/\Delta)) \cdot \mathrm{OPT}_k(\mathbf{X})$.

Finally, we combine the previous two steps to get bound (1). We argue that if the number of misses after $k$ rounds of $k$-means++ is less than $\Delta/2$, then almost all clusters are covered. Hence, we can apply bound (2) to $k' \leq \Delta/2$ uncovered clusters and $\Delta$ remaining rounds of $k$-means++ and get a $5(1 + 1/(2e))$ approximation. If the number of misses is greater than $\Delta/2$, then the result from the previous step yields an $O(\log(k/\Delta))$ approximation.

# 6  Analysis of $k$-means$\|$

In this section, we give a sketch of analysis for the $k$-means$\|$ algorithm. Specifically, we show upper bounds on the expected cost of the solution after $T$ rounds.

**Theorem 6.1.**  *The expected cost of the clustering returned by $k$-means$\|$ algorithm after $T$ rounds are upper bounded as follows:*

$$\text{for } \ell < k, \qquad \mathbb{E}\left[\mathrm{cost}_{T+1}(\mathbf{X})\right] \leq \left(e^{-\frac{\ell}{k}}\right)^T \mathbb{E}\left[\mathrm{cost}_1(\mathbf{X})\right] + \frac{5\mathrm{OPT}_k(\mathbf{X})}{1 - e^{-\frac{\ell}{k}}};$$

$$\text{for } \ell \geq k, \qquad \mathbb{E}\left[\mathrm{cost}_{T+1}(\mathbf{X})\right] \leq \left(\frac{k}{e\ell}\right)^T \mathbb{E}\left[\mathrm{cost}_1(\mathbf{X})\right] + \frac{5\mathrm{OPT}_k(\mathbf{X})}{1 - k/e\ell}.$$

**Remark:** For the second bound ($\ell \geq k$), the additive term $5\mathrm{OPT}_k(\mathbf{X})/(1 - k/(e\ell)) \leq 8\mathrm{OPT}_k(\mathbf{X})$.

The probability that a point is sampled by $k$-means$\|$ is strictly greater than the probability that it is sampled by $k$-means$\|_{\mathrm{Pois}}$ since $1 - e^{-\lambda} < \lambda$ for all $\lambda > 0$. Thus, for every round, we can couple $k$-means$\|_{\mathrm{Pois}}$ and $k$-means$\|$ so that each point sampled by $k$-means$\|_{\mathrm{Pois}}$ is also sampled by $k$-means$\|$. Thus, the expected cost returned by $k$-means$\|$ is at most the expected cost returned by $k$-means$\|_{\mathrm{Pois}}$. In the following analysis, we show an upper bound for the expected cost of the solution returned by $k$-means$\|_{\mathrm{Pois}}$.

As a thought experiment, consider a modified $k$-means$\|_{\mathrm{Pois}}$ algorithm. This algorithm is given the set $\mathbf{X}$, parameter $k$, and additionally the optimal solution $\mathcal{P} = \{P_1, \ldots, P_k\}$. Although this modified algorithm is useless in practice as we do not know the optimal solution in advance, it will be helpful for our analysis.

In every round $t$, the modified algorithm first draws independent Poisson random variables $Z_t(P_i) \sim \mathrm{Pois}(\lambda_t(P_i))$ for every cluster $i \in \{1, \ldots, k\}$ with rate $\lambda_t(P_i) = \sum_{x \in P_i} \lambda_t(x)$. Then, for each $i \in \{1, \ldots, k\}$, it samples $Z_t(P_i)$ points $x \in P_i$ with repetitions from $P_i$, picking every point $x$ with probability $\lambda_t(x)/\lambda_t(P_i)$ and adds them to the set of centers $C_t$. We assume that points in every set $C_t$ are ordered in the same way as they were chosen by this algorithm.

We claim that the distribution of the output sets $C_T$ of this algorithm is exactly the same as in the original $k$-means$\|_{\mathrm{Pois}}$ algorithm. Therefore, we can analyze the modified algorithm instead of $k$-means$\|_{\mathrm{Pois}}$, using the framework described in Sections 3.

**Lemma 6.2.**  *The sets $C_t$ in the original and modified $k$-means$\|_{\mathrm{Pois}}$ algorithms are identically distributed.*

*Proof.* Consider $|P_i|$ independent Poisson point processes $N_x(a)$ with rates $\lambda_t(x)$, where $x \in P_i$ (here, we use variable $a$ for time). Suppose we add a center $x$ at step $t$ of the algorithm if $N_x(t) \geq 1$. On the one hand, the probability that we choose $x$ is equal to $1 - e^{-\lambda_t(x)}$ which is exactly the

probability that $k$-means$\|_{\mathrm{Pois}}$ picks $x$ as a center at step $t$. On the other hand, the sum $N_{P_i} = \sum_{x \in P_i} N_x$ is a Poisson point process with rate $\lambda_t(P_i)$. Thus, the total number of jumps in the interval $[0,1]$ of processes $N_x$ with $x \in P_i$ is distributed as $Z_t(P_i)$. Moreover, the probability that $N_x$ jumps at time $a$ conditioned on the event that $N_{P_i}$ jumps at time $a$ is $\lambda_t(x)/\lambda_t(P_i)$. Thus, for every jump of $N_{P_i}$, we choose one random center $x$ with probability $\lambda_t(x)/\lambda_t(P_i)$. $\qquad\square$

**Lemma 6.3.** *For $k$-means$\|$ algorithm with parameter $\ell$, the following bounds hold:*

$$for\ \ell < k, \qquad \mathbb{E}\left[\mathrm{cost}_{t+1}(\mathbf{X})\right] \leq e^{-\frac{\ell}{k}} \cdot \mathbb{E}\left[\mathrm{cost}_t(\mathbf{X})\right] + 5\mathrm{OPT}_k(\mathbf{X});$$

$$for\ \ell \geq k, \qquad \mathbb{E}\left[\mathrm{cost}_{t+1}(\mathbf{X})\right] \leq \left(\frac{k}{e\ell}\right) \cdot \mathbb{E}\left[\mathrm{cost}_t(\mathbf{X})\right] + 5\mathrm{OPT}_k(\mathbf{X}).$$

*Proof.* Since the expected cost returned by $k$-means$\|$ is at most the expected cost returned by $k$-means$\|_{\mathrm{Pois}}$, we analyze the expected cost of the clustering after one step of $k$-means$\|_{\mathrm{Pois}}$.

If the algorithm covers cluster $P_i$ at round $t$, then at the next round, its uncovered cost equals $0$. The number of centers chosen in $P_i$ is determined by the Poisson random variable $Z_{t+1}(P_i)$. Hence, $P_i$ is uncovered at round $t+1$ only if $Z_{t+1}(P_i) = 0$. Since $U_t(P_i)$ is non-increasing in $t$ and $U_t(P_i) \leq \mathrm{cost}_t(P_i)$, we have

$$\mathbb{E}\left[U_{t+1}(P_i) \mid C_t\right] \leq \mathbb{P}\left[Z_{t+1}(P_i) = 0\right] U_t(P_i) \leq \exp\left(-\frac{\ell\,\mathrm{cost}_t(P_i)}{\mathrm{cost}_t(\mathbf{X})}\right)\mathrm{cost}_t(P_i).$$

Define two function: $f(x) = e^{-x} \cdot x$; and $g(x) = f(x)$ for $x \in [0,1]$ and $g(x) = e^{-1}$ for $x \in [1,\infty)$. Then,

$$\mathbb{E}\left[U_{t+1}(\mathbf{X}) \mid C_t\right] \leq \left(\frac{1}{k}\sum_{i=1}^{k} f\left(\frac{\ell\mathrm{cost}_t(P_i)}{\mathrm{cost}_t(\mathbf{X})}\right)\right)\frac{k\mathrm{cost}_t(\mathbf{X})}{\ell}.$$

Since $g(x) \leq f(x)$, and $g(x)$ is concave for $x \geq 0$, we have

$$\mathbb{E}\left[U_{t+1}(\mathbf{X}) \mid C_t\right] \leq \left(\frac{1}{k}\sum_{i=1}^{k} g\left(\frac{\ell\,\mathrm{cost}_t(P_i)}{\mathrm{cost}_t(\mathbf{X})}\right)\right)\frac{k\mathrm{cost}_t(\mathbf{X})}{\ell} \leq g\left(\frac{\ell}{k}\right)\frac{k\mathrm{cost}_t(\mathbf{X})}{\ell}.$$

Here, we use that $\sum_i \mathrm{cost}_t(P_i) = \mathrm{cost}_t(\mathbf{X})$.

Therefore, for $\ell \leq k$, we have $\mathbb{E}\left[U_{t+1}(\mathbf{X}) \mid C_t\right] \leq \left(e^{-\frac{\ell}{k}}\right)\mathrm{cost}_t(\mathbf{X})$; and for $\ell \geq k$, we have $\mathbb{E}\left[U_{t+1}(\mathbf{X}) \mid C_t\right] \leq \left(\frac{k}{e\ell}\right)\mathrm{cost}_t(\mathbf{X})$.

Similar to Corollary 4.4, the process $\widetilde{H}_t(P)$ for $k$-means$\|_{\mathrm{Pois}}$ is also a supermartingale, which implies $\mathbb{E}\left[H_{t+1}(\mathbf{X})\right] \leq 5\mathrm{OPT}_k(\mathbf{X})$. This concludes the proof. $\qquad\square$

*Proof of Theorem 6.1.* Applying the bound from Lemma 6.3 for $t$ times, we get the following results. For $\ell \leq k$,

$$\mathbb{E}\left[\mathrm{cost}_{t+1}(\mathbf{X})\right] \leq \left(e^{-\frac{\ell}{k}}\right)^t \mathbb{E}\left[\mathrm{cost}_1(\mathbf{X})\right] + 5\mathrm{OPT}_k(\mathbf{X})\eta_t,$$

where $\eta_t = \sum_{j=1}^{t} \left(e^{-\frac{\ell}{k}}\right)^{j-1} < \frac{1}{1-e^{-\frac{\ell}{k}}}$. For $\ell \geq k$,

$$\mathbb{E}\left[\mathrm{cost}_{t+1}(\mathbf{X})\right] \leq \left(\frac{k}{e\ell}\right)^t \mathbb{E}\left[\mathrm{cost}_1(\mathbf{X})\right] + 5\mathrm{OPT}_k(\mathbf{X})\eta_t,$$

where $\eta_t = \sum_{j=1}^{t} \left(\frac{k}{e\ell}\right)^{j-1} \leq \frac{1}{1-\frac{k}{e\ell}}$. $\qquad\square$

**Corollary 6.4.** *Consider a data set $\mathbf{X}$ with more than $k$ distinct points. Let $T = \ln(\mathbb{E}\left[\mathrm{cost}_1(\mathbf{X})\right]/\mathrm{OPT}_k(\mathbf{X}))$ and $\ell > k$. Then, after $T$ rounds of $k$-means$\|$, the expected cost of clustering $\mathbb{E}\left[\mathrm{cost}_T(\mathbf{X})\right]$ is at most $9\mathrm{OPT}_k(\mathbf{X})$.*

## Broader Impact

In this paper, we analyze very popular clustering algorithms and provide new approximation guarantees for them. We believe that our work gives new insights that will lead to development of better clustering algorithms.

## Acknowledgments and Disclosure of Funding

We would like to thank all the reviewers for their helpful comments. Konstantin Makarychev, Aravind Reddy, and Liren Shan were supported in part by NSF grants CCF-1955351 and HDR TRIPODS CCF-1934931. Aravind Reddy was also supported in part by NSF CCF-1637585.

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
