[Supplementary Material]

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

## Footnotes

[2]Recall, that $K(C_t)$ is a non-increasing stochastic process with $K(C_0) = k$.

[3]We use the bound $\Pr\{P \leq k\} \leq e^{-\lambda}(e\lambda/k)^k$, where $P$ is a Poisson random variable with parameter $\lambda$ and $k < \lambda$. See e.g., Theorem 5.4.2 in Mitzenmacher & Upfal (2017).

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

# Appendix

In this appendix, we first present our experiments, then give proofs omitted in the main part of the paper, and provide complimentary lower bounds.

## A    Experiments

In this section, we present plots that show that the performance of $k$-means$\|$ and "$k$-means++ with oversampling and pruning" algorithms are very similar in practice. Below, we compare the following algorithms on the datasets BioTest from KDD Cup 2004 (Elber, 2004) and COVTYPE from the UCI ML repository (Dua & Graff, 2017):

- Regular $k$-means++. The performance of this algorithm is shown with a solid black line on the plots below.
- $k$-means$\|$ without pruning. This algorithm samples $k$ centers using $k$-means$\|$ with $T = 5$ rounds and $\ell = k/T$.
- $k$-means$\|$. This algorithm first samples $5k$ centers using $k$-means$\|$ and then subsamples $k$ centers using $k$-means++. The performance of this algorithm is shown with a dashed blue line on the plots below.
- $k$-means++ with oversampling and pruning. This algorithm first samples $5k$ centers using $k$-means++ and then subsamples $k$ centers using $k$-means++. The performance of this algorithm is shown with a thin red line on the plots below.

For each $k = 5, 10, \cdots, 200$, we ran these algorithms for 50 iterations and took their average. We normalized all costs by dividing them by the cost of $k$-means++ with $k = 1000$ centers.

BioTest

BioTest

COVTYPE

COVTYPE

# B  Details for Preliminaries

For any set of points $\mathbf{Y} \subset \mathbb{R}^d$, let $\mu = \sum_{x \in \mathbf{Y}} x / |\mathbf{Y}|$ be the *centroid* of the cluster $\mathbf{Y}$. Then, the optimal cost of $\mathbf{Y}$ with one center,

$$\mathrm{OPT}_1(\mathbf{Y}) = \sum_{x \in \mathbf{Y}} \|x - \mu\|^2 = \frac{\sum_{(x,y) \in \mathbf{Y} \times \mathbf{Y}} \|x - y\|^2}{2 |\mathbf{Y}|}.$$

This is a well known formula which is often used for analyzing of $k$-means algorithms. For completeness, we give a proof below.

*Proof.* Consider any point $z \in \mathbb{R}^d$, then we have:

$$\begin{aligned}
\mathrm{cost}(\mathbf{Y}, \{z\}) = \sum_{x \in \mathbf{Y}} \|x - z\|^2 &= \sum_{x \in \mathbf{Y}} \|(x - \mu) + (\mu - z)\|^2 \\
&= \sum_{x \in \mathbf{Y}} \left( \|x - \mu\|^2 + \|\mu - z\|^2 + 2 \langle x - \mu, \mu - z \rangle \right) \\
&= \sum_{x \in \mathbf{Y}} \|x - \mu\|^2 + |\mathbf{Y}| \cdot \|\mu - z\|^2 + 2 \left\langle \sum_{x \in \mathbf{Y}} (x - \mu), \mu - z \right\rangle \\
&= \sum_{x \in \mathbf{Y}} \|x - \mu\|^2 + |\mathbf{Y}| \cdot \|\mu - z\|^2.
\end{aligned}$$

Thus, the optimal choice of $z$ to minimize $\mathrm{cost}(\mathbf{Y}, \{z\})$ is $\mu$ and $\mathrm{OPT}_1(\mathbf{Y}) = \sum_{x \in \mathbf{Y}} \|x - \mu\|^2$.

$$\sum_{x \in \mathbf{Y}} \|x - \mu\|^2 = \sum_{x \in \mathbf{Y}} \langle x - \mu, x - \mu \rangle = \sum_{x \in \mathbf{Y}} \langle x, x - \mu \rangle$$

$$= \sum_{x \in \mathbf{Y}} \left\langle x, x - \sum_{y \in \mathbf{Y}} \frac{y}{|\mathbf{Y}|} \right\rangle = \frac{1}{|\mathbf{Y}|} \sum_{(x,y) \in \mathbf{Y} \times \mathbf{Y}} \langle x, x - y \rangle$$

$$= \frac{1}{2|\mathbf{Y}|} \left( \sum_{(x,y) \in \mathbf{Y} \times \mathbf{Y}} \langle x, x - y \rangle + \sum_{(x,y) \in \mathbf{Y} \times \mathbf{Y}} \langle y, y - x \rangle \right)$$

$$= \frac{\sum_{(x,y) \in \mathbf{Y} \times \mathbf{Y}} \|x - y\|^2}{2|\mathbf{Y}|}.$$

$\square$

## C   Details for Section 4

**Lemma 4.2.**   *For any $x, y \in P$, we have $f(x, y) \le 5 \min \{\mathrm{cost}(x, C), \mathrm{cost}(y, C)\} \|x - y\|^2$.*

where $f(x, y) = \mathrm{cost}(x, C) \cdot \min \{\|x - y\|^2, \mathrm{cost}(y, C)\} + \mathrm{cost}(y, C) \cdot \min \{\|x - y\|^2, \mathrm{cost}(x, C)\}$

*Proof.* Since $f(x, y)$ is a symmetric function with respect to $x$ and $y$, we may assume without loss of generality that $\mathrm{cost}(x, C) \le \mathrm{cost}(y, C)$. Then, we need to show that $f(x, y) \le 5\mathrm{cost}(x, C)\|x - y\|^2$. Consider three cases.

**Case 1:** If $\mathrm{cost}(x, C) \le \mathrm{cost}(y, C) \le \|x - y\|^2$, then

$$f(x, y) = 2\mathrm{cost}(x, C)\mathrm{cost}(y, C) \le 2\mathrm{cost}(x, C)\|x - y\|^2.$$

**Case 2:** If $\mathrm{cost}(x, C) \le \|x - y\|^2 \le \mathrm{cost}(y, C)$, then

$$f(x, y) = \mathrm{cost}(x, C)\|x - y\|^2 + \mathrm{cost}(y, C)\mathrm{cost}(x, C).$$

By the triangle inequality, we have

$$\mathrm{cost}(y, C) \le \left( \sqrt{\mathrm{cost}(x, C)} + \|x - y\| \right)^2 \le 4\|x - y\|^2.$$

Thus, $f(x, y) \le 5\mathrm{cost}(x, C)\|x - y\|^2$.

**Case 3:** If $\|x - y\|^2 \le \mathrm{cost}(x, C) \le \mathrm{cost}(y, C)$, then

$$f(x, y) = (\mathrm{cost}(x, C) + \mathrm{cost}(y, C)) \|x - y\|^2.$$

By the triangle inequality,

$$\mathrm{cost}(y, C) \le \left( \sqrt{\mathrm{cost}(x, C)} + \|x - y\| \right)^2 \le 4\mathrm{cost}(x, C).$$

Thus, we have $f(x, y) \le 5\mathrm{cost}(x, C)\|x - y\|^2$.

In all cases, the desired inequality holds. This concludes the proof of Lemma 4.2.   $\square$

**Lemma C.1.** *There exists a maximizer $\phi^{**}$ of $F(\phi)$ in the region $\{\phi \ge 0\}$ such that $\phi \in \{0, 1\}^P$.*

*Proof.* Let $m = |P|$ be the size of the cluster $P$ and $\Pi$ be the set of all bisections or permutations $\pi : \{1, \ldots, m\} \to P$. Partition the set $\{\phi \ge 0\}$ into $m!$ regions ("cones over order polytopes"):

$$\{\phi : \phi \ge 0\} = \cup_{\pi \in \Pi} O_\pi,$$

where $O_\pi = \{\phi : 0 \le \phi_{\pi(1)} \le \phi_{\pi(2)} \le \cdots \le \phi_{\pi(m)}\}$. We show that for every $\pi \in \Pi$, there exists a maximizer $\phi^{**}$ of $F(\phi)$ in the region $O_\pi$, such that $\phi^{**} \in \{0, 1\}^P$. Therefore, there exists a global maximizer $\phi^{**}$ that belongs $\{0, 1\}^P$

Fix a $\pi \in \Pi$. Denote by $V$ the hyperplane $\{\phi : \sum_{x \in P} \phi_x = 1\}$. Observe that $F$ is a scale invariant function i.e., $F(\phi) = F(\lambda\phi)$ for every $\lambda > 0$. Thus, for every $\phi \in O_\pi$, there exists a $\phi' \in O_\pi \cap V$ (namely, $\phi' = \phi/(\sum_{x \in P} \phi_x)$) such that $F(\phi') = F(\phi)$. Hence, $\max\{F(\phi) : \phi \in O_\pi\} = \max\{F(\phi) : \phi \in O_\pi \cap V\}$. Note that for $\phi \in V$, the denominator of (3) equals 2, and for $\phi \in O_\pi$, the numerator of (3) is a linear function of $\phi$. Therefore, $F(\phi)$ is a linear function in the convex set $O_\pi \cap V$. Consequently, one of the maximizers of $F$ must be an extreme point of $O_\pi \cap V$.

The polytope $O_\pi \cap V$ is defined by $m$ inequalities and one equality. Thus, for every extreme point $\phi$ of this polytope, all inequalities $\phi_{\pi(i)} \leq \phi_{\pi(i+1)}$ but one must be tight. In other words, for some $j < m$, we have

$$0 = \phi_{\pi(1)} = \cdots = \phi_{\pi(j)} < \phi_{\pi(j+1)} = \cdots = \phi_{\pi(m)}. \tag{4}$$

Therefore, there exists a maximizer $\phi$ of $F(\phi)$ in $O_\pi \cap V$ satisfying (4) for some $j$. After rescaling $\phi$ – multiplying all coordinates of $\phi$ by $(m-j)$ – we obtain a vector $\phi^{**}$ whose first $j$ coordinates $\phi_{\pi(1)}^{**}, \ldots, \phi_{\pi(j)}^{**}$ are zeroes and the last $m - j$ coordinates $\phi_{\pi(j+1)}^{**}, \ldots, \phi_{\pi(m)}^{**}$ are ones. Thus, $\phi^{**} \in \{0,1\}^P$. Since $F$ is rescaling invariant, $F(\phi^{**}) = F(\phi)$. This concludes the proof. $\square$

# D   Bi-criteria approximation for $k$-means++

In this section, we analyze the bi-criteria $k$-means++ algorithm and prove Theorem 5.1. To this end, we establish the first and second bounds from Theorem 5.1 on the expected cost of the clustering after $k + \Delta$ rounds of $k$-means. We will start with the second bound.

## D.1   Bi-criteria bound for large $\Delta$

**Lemma D.1.** *The following bi-criteria bound holds*

$$\mathbb{E}\left[\mathrm{cost}_{k+\Delta}\left(\mathbf{X}\right)\right] \leq 5 \left(1 + \frac{k}{e\left(\Delta - 1\right)}\right) \mathrm{OPT}_k(\mathbf{X}).$$

Consider the discrete time Markov chain $C_t$ associated with $k$-means++ algorithm (see Section 3). Let $P \in \mathcal{P}$ be an arbitrary cluster in the optimal solution. Partition all states of the Markov chain into $k + \Delta$ disjoint groups $\mathcal{M}_0, \mathcal{M}_1, \cdots, \mathcal{M}_{k+\Delta-1}$ and $\mathcal{H}$. Each set $\mathcal{M}_i$ contains all states $C$ with $i$ misses that do not cover $P$: $\mathcal{M}_i = \{C : M(C) = i, P \cap C = \varnothing\}$. The set $\mathcal{H}$ contains all states $C$ that cover $P$: $\mathcal{H} = \{C : P \cap C \neq \varnothing\}$.

We now define a new Markov chain $X_t$. To this end, we first expand the set of states $\{C\}$. For every state $C$ of the process $C_t$, we create two additional "virtual" states $C^a$ and $C^b$. Then, we let $X_{2t} = C_t$ for every even step $2t$, and

$$X_{2t+1} = \begin{cases} C_t^a, & \text{if } C_{t+1} \in \mathcal{M}_i \\ C_t^b, & \text{if } C_{t+1} \in \mathcal{M}_{i+1} \cup \mathcal{H}. \end{cases}$$

for every odd step $2t + 1$. We stop $X_t$ when $C_t$ stops or when $C_t$ hits the set $\mathcal{H}$ (i.e., $C_t \in \mathcal{H}$). Loosely speaking, $X_t$ follows Markov chain $C_t$ but makes additional intermediate stops. When $C_t$ moves from one state in $\mathcal{M}_i$ to another state in $\mathcal{M}_i$, $X_{2t+1}$ stops in $C_t^a$; and when $C_t$ moves from a state in $\mathcal{M}_i$ to a state in $\mathcal{M}_{i+1}$ or $\mathcal{H}$, $X_{2t+1}$ stops in $C_t^b$.

Write transition probabilities for $X_t$:

$$\mathbb{P}\left[X_{2t+1} = C^a \mid X_{2t} = C\right] = \frac{U(\mathbf{X}, C) - U(P, C)}{\mathrm{cost}(\mathbf{X}, C)},$$

$$\mathbb{P}\left[X_{2t+1} = C^b \mid X_{2t} = C\right] = \frac{U(P, C) + H(\mathbf{X}, C)}{\mathrm{cost}(\mathbf{X}, C)},$$

and for all $C \in \mathcal{M}_i$ and $C' = C \cup \{x\} \in \mathcal{M}_i$,

$$\mathbb{P}\left[X_{2t+2} = C' \mid X_{2t+1} = C^a\right] = \frac{\mathrm{cost}(x, C)}{U(\mathbf{X}, C) - U(P, C)},$$

for all $C \in \mathcal{M}_i$ and $C' = C \cup \{x\} \in \mathcal{M}_{i+1} \cup \mathcal{H}$,

$$\mathbb{P}\left[X_{2t+2} = C' \mid X_{2t+1} = C^b\right] = \frac{\mathrm{cost}(x, C)}{U(P, C) + H(\mathbf{X}, C)}.$$

Above, $U(\mathbf{X}, C) - U(P, C)$ is the cost of points in all uncovered clusters except for $P$. If we pick a center from these clusters, we will necessarily cover a new cluster, and therefore $X_{2t+2}$ will stay in $\mathcal{M}_i$. Similarly, $U(P, C) + H(\mathbf{X}, C)$ is the cost of all covered clusters plus the cost of $P$. If we pick a center from these clusters, then $X_{2t+2}$ will move to $\mathcal{M}_{i+1}$ or $\mathcal{H}$.

Define another Markov chain $\{Y_t\}$. The transition probabilities of $\{Y_t\}$ are the same as the transition probabilities of $X_t$ except $Y$ never visits states in $\mathcal{H}$ and therefore for $C \in \mathcal{M}_i$ and $C' = C \cup \{x\} \in \mathcal{M}_{i+1}$, we have

$$\mathbb{P}\left[ Y_{2t+2} = C' \mid Y_{2t+1} = C^b \right] = \frac{\text{cost}(x, C)}{H(\mathbf{X}, C)}.$$

We now prove a lemma that relates probabilities of visiting states by $X_t$ and $Y_t$.

**Lemma D.2.** *For every $t \le k + \Delta$ and states $C' \in \mathcal{M}_i$, $C'' \in \mathcal{M}_\Delta$, we have*

$$\frac{\mathbb{P}\left[ C'' \in \{X_j\} \mid X_{2t} = C' \right]}{\mathbb{P}\left[ C'' \in \{Y_j\} \mid Y_{2t} = C' \right]} \le \left( \frac{\widetilde{H}(\mathbf{X}, C'')}{\widetilde{H}(\mathbf{X}, C'') + U(P, C'')} \right)^{\Delta - i}$$

*where $\{C'' \in \{X_j\}\}$ and $\{C'' \in \{Y_j\}\}$ denote the events $X$ visits $C''$ and $Y$ visits $C''$, respectively.*

*Proof.* Consider the unique path $p$ from $C'$ to $C''$ in the state space of $X$ (note that the transition graphs for $X$ and $Y$ are directed trees). The probability of transitioning from $C'$ to $C''$ for $X$ and $Y$ equals the product of respective transition probabilities for every edge on the path. Recall that transitions probabilities for $X$ and $Y$ are the same for all states but $C^b$, where $C \in \cup_j \mathcal{M}_j$. The number of such states on the path $p$ is equal to the number transitions from $\mathcal{M}_j$ to $\mathcal{M}_{j+1}$, since $X$ and $Y$ can get from $\mathcal{M}_j$ to $\mathcal{M}_{j+1}$ only through a state $C^b$ on the boundary of $\mathcal{M}_j$ and $\mathcal{M}_{j+1}$. The number of transitions from $\mathcal{M}_j$ to $\mathcal{M}_{j+1}$ equals $\Delta - i$. For each state $C^b$ on the path, the ratio of transition probabilities from $C^b$ to the next state $C \cup \{x\}$ for Markov chains $X$ and $Y$ equals

$$\frac{H(\mathbf{X}, C)}{U(P, C) + H(\mathbf{X}, C)} \le \frac{\widetilde{H}(\mathbf{X}, C'')}{U(P, C'') + \widetilde{H}(\mathbf{X}, C'')},$$

here we used that (a) $U(P, C) \ge U(P, C'')$ since $U_t(P)$ is a non-increasing process; and (b) $H(P, C) \le \widetilde{H}(P, C'')$ since $H_t(P) \le \widetilde{H}_{t'}(P)$ if $t \le t'$ (see Section 3). $\qquad \square$

We now prove an analog of Corollary 4.4 for $\widetilde{H}(\mathbf{X}, Y_j)$.

**Lemma D.3.** $\widetilde{H}(\mathbf{X}, Y_t)$ *is a supermartingale.*

*Proof.* If $Y_j = C$, then $Y_{j+1}$ can only be in $\left\{ C^a, C^b \right\}$. Since $\widetilde{H}(\mathbf{X}, C^a) = \widetilde{H}(\mathbf{X}, C^b) = \widetilde{H}(\mathbf{X}, C)$, we have $\mathbb{E}\left[ \widetilde{H}(\mathbf{X}, Y_{j+1}) \mid Y_j = C \right] = \widetilde{H}(\mathbf{X}, Y_j)$.

If $Y_j = C^a$, then $Y_{j+1} = C'$ where the new center $c$ should be in uncovered clusters with respect to $C_t$.

$$\mathbb{E}\left[ H(P', Y_{j+1}) \mid Y_j = C^a, c \in P' \right] \le 5\text{OPT}_1(P'),$$

which implies

$$\mathbb{E}\left[ \widetilde{H}(P', Y_{j+1}) \mid Y_j = C^a, c \in P' \right] \le \widetilde{H}(P', Y_j).$$

Therefore, we have

$$\mathbb{E}\left[ \widetilde{H}(\mathbf{X}, Y_{j+1}) \mid Y_j = C^a \right] \le \widetilde{H}(\mathbf{X}, Y_j).$$

If $Y_j = C^b$, then for any possible state $C'$ of $Y_{j+1}$, the new center should be in covered clusters with respect to $C$. By definition, we must have $\widetilde{H}(\mathbf{X}, C') = \widetilde{H}(\mathbf{X}, C) = \widetilde{H}(\mathbf{X}, C^b)$. Thus, it holds that $\mathbb{E}\left[ \widetilde{H}(\mathbf{X}, Y_{j+1}) \mid Y_j = C^b \right] = \widetilde{H}(\mathbf{X}, Y_j)$.

Combining all these cases, we get $\left\{ \widetilde{H}(\mathbf{X}, Y_j) \right\}$ is a supermartingale. $\qquad \square$

We now use Lemma D.2 and Lemma D.3 to bound the expected uncovered cost of $P$ after $k + \Delta$ rounds of $k$-means++.

**Lemma D.4.** *For any cluster $P \in \mathcal{P}$ and $t \leq k + \Delta$, we have*

$$\mathbb{E}\left[U_{k+\Delta}(P) \mid C_t\right] \leq \frac{\widetilde{H}_t(\mathbf{X})}{e(\Delta - M(C_t) - 1)}.$$

*Proof.* Since $k$-means++ samples $k + \Delta$ centers and the total number of clusters in the optimal solution $\mathcal{P}$ is $k$, $k$-means++ must make $\Delta$ misses. Hence, the process $\{X_t\}$ which follows $k$-means++ must either visit a state in $\mathcal{M}_{\geq\Delta}$ or stop in $\mathcal{H}$ (recall that we stop process $X_t$ if it reaches $\mathcal{H}$).

If $\{X_t\}$ stops in group $\mathcal{H}$, then the cluster $P$ is covered which means that $U_{k+\Delta}(P) = 0$. Let $\partial\mathcal{M}_\Delta$ be the frontier of $\mathcal{M}_\Delta$ i.e., the states that $X_t$ visits first when it reaches $\mathcal{M}_\Delta$ (recall that the transition graph of $X_t$ is a tree). The expected cost $\mathbb{E}\left[U_{k+\Delta}(P) \mid C_t\right]$ is upper bounded by the expected uncovered cost of $P$ at time when $C_t$ reaches $\partial\mathcal{M}_\Delta$. Thus,

$$\mathbb{E}\left[U_{k+\Delta}(P) \mid C_t\right] \leq \sum_{C \in \partial\mathcal{M}_\Delta} \mathbb{P}\left[C \in \{X_j\} \mid C_t\right] U(P, C).$$

Observe that by Lemma D.2, for any $C \in \partial\mathcal{M}_\Delta$, we have

$$\mathbb{P}\left[C \in \{X_j\} \mid C_t\right] U(P, C) \leq \mathbb{P}\left[C \in \{Y_j\} \mid C_t\right] \left(\frac{\widetilde{H}(\mathbf{X}, C)}{\widetilde{H}(\mathbf{X}, C) + U(P, C)}\right)^{\Delta'} U(P, C).$$

Let $f(x) = x(1/(1 + x))^{\Delta'}$. Then, $f(x)$ is maximized at $x = 1/(\Delta' - 1)$ and the maximum value $f(1/(\Delta' - 1)) = 1/(e(\Delta' - 1))$. Therefore, for every $C \in \partial\mathcal{M}_\Delta$, we have

$$\mathbb{P}\left[C \in \{X_j\} \mid C_t\right] U(P, C) \leq \mathbb{P}\left[C \in \{Y_j\} \mid C_t\right] f\left(\frac{U(P, C)}{\widetilde{H}(\mathbf{X}, C)}\right) \widetilde{H}(\mathbf{X}, C)$$

$$\leq \mathbb{P}\left[C \in \{Y_j\} \mid C_t\right] \frac{\widetilde{H}(\mathbf{X}, C)}{e(\Delta' - 1)}.$$

Let $\tau = \min\{j : Y_j \in \partial\mathcal{M}_\Delta\}$ be the stopping time when $Y_j$ first visits $\partial\mathcal{M}_\Delta$. We get

$$\sum_{C \in \partial\mathcal{M}_\Delta} \mathbb{P}\left[C \in \{Y_j\} \mid C_t\right] \widetilde{H}(\mathbf{X}, C) = \mathbb{E}\left[\widetilde{H}(\mathbf{X}, Y_\tau) \mid C_t\right].$$

By Lemma D.3, $\widetilde{H}(\mathbf{X}, Y_j)$ is a supermartingale. Thus, by the optional stopping theorem, $\mathbb{E}\left[\widetilde{H}(\mathbf{X}, Y_\tau) \mid C_t\right] \leq \widetilde{H}(\mathbf{X}, C_t)$. Therefore, we have

$$\mathbb{E}\left[U_{k+\Delta}(P) \mid C_t\right] \leq \frac{\widetilde{H}_t(\mathbf{X})}{e(\Delta - M(C_t) - 1)},$$

This concludes the proof. $\qquad\square$

We now add up bounds from Lemma D.4 with $t = 0$ for all clusters $P \in \mathcal{P}$ and obtain Lemma D.1.

## D.2 Bi-criteria bound for small $\Delta$

In this section, we give another bi-criteria approximation guarantee for $k$-means++.

**Lemma D.5.** *Let $\text{cost}_{k+\Delta}(\mathbf{X})$ be the cost of the the clustering resulting from sampling $k + \Delta$ centers according to the $k$-means++ algorithm (for $\Delta \in \{1, \ldots, 2k\}$). Then,*

$$\mathbb{E}\left[\text{cost}_{k+\Delta}(X)\right] \leq 5\left(2 + \frac{1}{2e} + \ln\frac{2k}{\Delta}\right) \text{OPT}_k(X).$$

*Proof.* Consider $k$-means++ clustering algorithm and the corresponding random process $C_t$. Fix a $\kappa \in \{1, \ldots, k\}$. Let $\tau$ be the first iteration[2] (stopping time) when $K(C_\tau) \leq \kappa$ if $K(C_k) \leq \kappa$; and $\tau = k$, otherwise. We refer the reader to Section 3 for definitions of $M(C_t)$, $U_t(X) = U(X, C_t)$, and $K(C_t)$.

We separately analyze the cost of uncovered clusters after the first $\tau$ steps and the last $k' - \tau$ steps, where $k' = k + \Delta$ is the total number of centers chosen by $k$-means++.

The first step of our proof follows the analysis of $k$-means++ by Dasgupta (2013), and by Arthur and Vassilvitskii (2007). Define a potential function $\Psi$ (see Dasgupta 2013):

$$\Psi_t := \frac{M(C_t)U(X, C_t)}{K(C_t)}.$$

If $K(C_t) = 0$, then $M(C_t)$ and $U(X, C_t)$ must be 0 and we let $\Psi_t = 0$

We use the following result by Dasgupta (2013) to estimate $\mathbb{E}[\Psi_\tau(X)]$ in Lemma D.7.

**Lemma D.6** (Dasgupta (2013)). *For any $0 \leq t \leq k$, we have*

$$\mathbb{E}\left[\Psi_{t+1} - \Psi_t \mid C_t\right] \leq \frac{H(X, C_t)}{K(C_t)}.$$

**Lemma D.7.** *Then, the following bound holds:*

$$\mathbb{E}[\Psi_\tau(X)] \leq 5\left(1 + \ln\left(\frac{k}{\kappa + 1}\right)\right)\mathrm{OPT}_k(X).$$

*Proof.* Note that $\Psi_1 = 0$ as $M(C_1) = 0$. Thus,

$$\mathbb{E}[\Psi_\tau] \leq \sum_{t=1}^{\tau-1} \mathbb{E}\left[\Psi_{t+1} - \Psi_t\right] \leq \mathbb{E}\left[\sum_{t=1}^{\tau-1} \frac{H(X, C_t)}{K(C_t)}\right].$$

Using the inequality $H(X, C_t) \leq \widetilde{H}_k(X)$ (see Section 3), we get:

$$\mathbb{E}[\Psi_\tau] \leq \mathbb{E}\left[\sum_{t=1}^{\tau-1} \frac{\widetilde{H}_k(X)}{K(C_t)}\right] \leq \mathbb{E}\left[\widetilde{H}_k(X) \cdot \sum_{t=1}^{\tau-1} \frac{1}{K(C_t)}\right].$$

Observe that $K(C_1), \ldots, K(C_{\tau-1})$ is a non-increasing sequence in which two consecutive terms are either equal or $K(C_{i+1}) = K(C_i) - 1$. Moreover, $K(C_1) = k$ and $K(C_{\tau-1}) > \kappa$. Therefore, by Lemma D.8 (see below), for every realization $C_0, C_1, \ldots, C_\tau$, we have:

$$\sum_{t=1}^{\tau-1} \frac{1}{K(C_t)} \leq 1 + \log {k}/{(\kappa+1)}.$$

Thus, $\mathbb{E}[\Psi_\tau] \leq (1 + \log {k}/{(\kappa+1)})\mathbb{E}[\widetilde{H}_k(X)] \leq 5(1 + \log {k}/{(\kappa+1)})\,\mathrm{OPT}_k(X)$. This concludes the proof. $\qquad\square$

Let $\kappa = \lfloor(\Delta - 1)/2\rfloor$. By Lemma D.7, we have

$$\mathbb{E}\left[\frac{M(C_\tau)U_\tau(X)}{K(C_\tau)}\right] \leq 5\left(1 + \ln\frac{2k}{\Delta}\right)\mathrm{OPT}_k(X).$$

Since $U_t(X)$ is a non-increasing stochastic process, we have $\mathbb{E}[U_{k+\Delta}(X)] \leq \mathbb{E}[U_\tau(X)]$. Thus,

$$\mathbb{E}\left[\frac{M(C_\tau)}{K(C_\tau)} \cdot U_{k+\Delta}(X)\right] \leq 5\left(1 + \ln\frac{2k}{\Delta}\right)\mathrm{OPT}_k(X).$$

Our goal is to bound $\mathbb{E}[U_{k'}(X)]$. Write,

$$\mathbb{E}[U_{k'}(X)] = \mathbb{E}\left[\frac{M(C_\tau)}{K(C_\tau)} \cdot U_{k'}(X)\right] + \mathbb{E}\left[\frac{K(C_\tau) - M(C_\tau)}{K(C_\tau)} \cdot U_{k'}(X)\right].$$

The first term on the right hand side is upper bounded by $5\left(1 + \ln \frac{2k}{\Delta}\right)\mathrm{OPT}_k(X)$. We now estimate the second term, which we denote by $(*)$.

Note that $K(C_t) - M(C_t) = k - t$, since the number of uncovered clusters after $t$ steps of $k$-means++ equals the number of misses plus the number of steps remaining. Particularly, if $\tau = k$, we have $K(C_\tau) - M(C_\tau) = K(C_k) - M(C_k) = 0$. Consequently, if $\tau = k$, then the second term $(*)$ equals 0. Thus, we only need to consider the case, when $\tau < k$. Note that in this case $K(C_\tau) = \kappa$. By Lemma D.1 (applied to all uncovered clusters), we have

$$\mathbb{E}[U_{k'}(X) \mid C_\tau, \tau] \le \frac{K(C_\tau)}{e(\Delta' - 1)}\widetilde{H}_\tau(X),$$

where $\Delta' = \Delta - M(C_\tau)$.

Thus,

$$\mathbb{E}\left[\frac{K(C_\tau) - M(C_\tau)}{K(C_\tau)} \cdot U_{k'}(X) \mid C_\tau, \tau\right] \le \frac{K(C_\tau) - M(C_\tau)}{K(C_\tau)} \cdot \frac{K(C_\tau)}{e(\Delta' - 1)} \cdot \widetilde{H}_\tau(X) = (**).$$

Plugging in $K(C_\tau) = \kappa$ and the expression for $\Delta'$ (see above), and using that $\kappa \le (\Delta - 1)/2$, we get

$$(**) = \frac{\kappa - M(C_\tau)}{e(\Delta - M(C_\tau) - 1)} \cdot \widetilde{H}_\tau(X) \le \frac{1}{2e}\widetilde{H}_\tau(X).$$

Finally, taking the expectation over all $C_\tau$, we obtain the bound

$$\mathbb{E}\left[\frac{K(C_\tau) - M(C_\tau)}{K(C_\tau)} \cdot U_{k'}(X)\right] \le \frac{5\mathrm{OPT}_1(X)}{2e}.$$

Thus, $\mathbb{E}[U_{k'}(X)] \le 5(1 + {}^1\!/2e + \ln {}^{2k}\!/\Delta)\mathrm{OPT}_k(X)$. Therefore, $\mathbb{E}[\mathrm{cost}_{k'}(X)] = \mathbb{E}[H_{k'}(X)] + U_{k'}(X) \le 5\left(2 + \frac{1}{2e} + \ln \frac{2k}{\Delta}\right)\mathrm{OPT}_k(X)$. $\qquad\square$

We now prove Lemma D.8.

**Lemma D.8.** *For any $t \le k$ integers $a_1 \ge a_2 \ge \cdots \ge a_t$ such that $a_1 = k$, $a_t > \kappa$ and $a_i - a_{i+1} \in \{0, 1\}$ for all $1 \le i < t$, the following inequality holds*

$$\sum_{i=1}^{t} \frac{1}{a_i} \le 1 + \log\left(\frac{k}{\kappa + 1}\right).$$

*Proof.* It is easy to see that the sum is maximized when $t = k$, and the sequence $a_1, \ldots, a_k$ is as follows:

$$\underbrace{\frac{1}{k}, \frac{1}{k-1}, \ldots, \frac{1}{\kappa + 2}}_{(k - (\kappa + 1)) \text{ terms}}, \underbrace{\frac{1}{\kappa + 1}, \ldots, \frac{1}{\kappa + 1}}_{(\kappa + 1) \text{ terms}}.$$

The sum of the first $(k - (\kappa + 1))$ terms is upper bounded by

$$\int_{1/(\kappa + 1)}^{1/k} \frac{1}{x} \, dx = \ln \frac{k}{\kappa + 1}.$$

The sum of the last $(\kappa + 1)$ terms is 1. $\qquad\square$

# E   Lower bounds

## E.1   Lower bound on the cost of covered clusters

We show the following lower bound on the expected cost of a covered cluster in $k$-means++. Therefore, the 5-approximation in Lemma 4.1 is tight.

**Theorem E.1.** *For any $\varepsilon > 0$, there exists an instance of $k$-means such that for a set $P \in \mathbf{X}$ and a set of centers $C \in \mathbb{R}^d$, if a new center $c$ is sampled from $P$ with probability $\Pr(c = x) = \mathrm{cost}(x, C)/\mathrm{cost}(P, C)$, then*

$$\mathbb{E}_c\left[\mathrm{cost}(P, C \cup \{c\})\right] \geq (5 - \varepsilon)\mathrm{OPT}_1(P).$$

*Proof.* Consider the following one dimensional example, where $P$ contains $t$ points at $0$ and one point at $1$, and the closest center already chosen in $C$ to $P$ is at $-1$.

The new center $c$ will be chosen at $0$ with probability $\frac{t}{t+4}$, and at $1$ with probability $\frac{4}{t+4}$. Then, the expected cost of $P$ is

$$\mathbb{E}_c\left[\mathrm{cost}(P, C \cup \{c\})\right] = 1 \cdot \frac{t}{t+4} + t \cdot \frac{4}{t+4} = \frac{5t}{t+4};$$

and the optimal cost of $P$ is $\mathrm{OPT}_1(P) \leq 1$. Thus, by choosing $t \geq 4(5 - \varepsilon)/\varepsilon$, we have

$$\mathbb{E}_c\left[\mathrm{cost}(P, C \cup \{c\})\right] \geq (5 - \varepsilon)\mathrm{OPT}_1(P).$$

$\square$

## E.2   Lower bound on the bi-criteria approximation

In this section, we show that the bi-criteria approximation bound of $O(\ln \frac{k}{\Delta})$ is tight up to constant factor. Our proof follows the approach by Brunsch & Röglin (2013). We show the following theorem.

**Theorem E.2.** *For every $k > 1$ and $\Delta \leq k$, there exists an instance $\mathbf{X}$ of $k$-means such that the bi-criteria $k$-means++ algorithm with $k + \Delta$ centers returns a solution of cost greater than*

$$\frac{1}{8}\log\frac{k}{\Delta} \cdot \mathrm{OPT}_k(\mathbf{X})$$

*with probability at least $1 - e^{-\sqrt{k}/2}$.*

**Remark:** This implies that the expected cost of bi-criteria $k$-means with $k + \Delta$ centers is at least

$$\frac{1 - e^{-\sqrt{k}/2}}{8} \cdot \log\frac{k}{\Delta} \cdot \mathrm{OPT}_k(\mathbf{X}).$$

*Proof.* For every $k$ and $\Delta \geq \sqrt{k}$, we consider the following instance. The first cluster is a scaled version of the standard simplex with $N \gg k$ vertices centered at the origin, which is called the heavy cluster. The length of the edges in this simplex is $1/\sqrt{N-1}$. Each of the remaining $k - 1$ clusters contains a single point on $k - 1$ axes, which are called light clusters. These clusters are located at distance $\sqrt{\alpha}$ from the center of the heavy cluster and $\sqrt{2\alpha}$ from each other, where $\alpha = \frac{\ln(k/\Delta)}{4\Delta}$.

For the sake of analysis, let us run $k$-means++ till we cover all clusters. At the first step, the $k$-means++ algorithm almost certainly selects a center from the heavy cluster since $N \gg k$. Then, at each step, the algorithm can select a center either from one of uncovered light clusters or from the heavy cluster. In the former case, we say that the algorithm hits a light cluster, and in the latter case we say that the algorithm misses a light cluster. Below, we show that with high probability the algorithm makes at least $2\Delta$ misses before it covers all but $\Delta$ light clusters.

**Lemma E.3.** *Let $\Delta \geq \sqrt{k}$. By the time the $k$-means++ algorithm covers all but $\Delta$ light clusters, it makes greater than $2\Delta$ misses with probability at least $1 - e^{-\sqrt{k}/2}$.*

*Proof sketch.* Let $\varepsilon = 1/\sqrt{N}$. Observe that $k$-means++ almost certainly covers all clusters in $\varepsilon N$ steps (since $N \gg k$). So in the rest of this proof sketch, we assume that the number chosen centers is at most $\varepsilon N$ and, consequently, at least $(1 - \varepsilon)N$ points in the heavy cluster are not selected as centers. Hence, the cost of the heavy cluster is at least $1 - \varepsilon$.

Consider a step of the algorithm when exactly $u$ light clusters remain uncovered. At this step, the total cost of all light clusters is $\alpha u$ (we assume for simplicity that distance between the light clusters and the closest chosen center in the heavy cluster is the same as the distance to the origin). The cost of the heavy cluster is at least $1 - \varepsilon$. The probability that the algorithm chooses a center from the heavy cluster and thus misses a light cluster is at least $(1 - \varepsilon)/(1 + \alpha u)$.

Define random variables $\{X_u\}$ as follows. Let $X_u = 1$ if the algorithm misses a cluster at least once when the number of uncovered light clusters is $u$; and let $X_u = 0$, otherwise. Then, $\{X_u\}$ are independent Bernoulli random variables. For each $u$, we have $\mathbb{P}[X_u = 1] \geq (1 - \varepsilon)/(1 + \alpha u)$.

Observe that the total number of misses is lower bounded by $\sum_{u=\Delta}^{k-1} X_u$. Then, we have

$$\mathbb{E}\left[\sum_{u=\Delta}^{k-1} X_u\right] \geq (1 - \varepsilon) \sum_{u=\Delta}^{k-1} \frac{1}{1 + \alpha u} \geq (1 - \varepsilon) \int_{\Delta}^{k} \frac{du}{1 + \alpha u}$$

$$= (1 - \varepsilon)\alpha^{-1} \ln \frac{1 + \alpha k}{1 + \alpha \Delta}$$

$$\geq (1 - \varepsilon)\alpha^{-1} \ln \frac{k}{\Delta} = 4(1 - \varepsilon)\Delta.$$

Let $\mu = \mathbb{E}\left[\sum_{u=\Delta}^{k-1} X_u\right] \geq 4(1 - \varepsilon)\Delta$. By the Chernoff bound for Bernoulli random variables, we have

$$\mathbb{P}\left[\sum_{u=\Delta}^{k} X_u \leq 2\Delta\right] \leq e^{-\mu} \left(\frac{e\mu}{2\Delta}\right)^{2\Delta}.$$

Since $f(x) = e^{-x}(\frac{ex}{2\Delta})^{2\Delta}$ is a monotone decreasing function for $x \geq 2\Delta$, we have

$$\mathbb{P}\left[\sum_{u=\Delta}^{k} X_u \leq 2\Delta\right] \leq e^{-(2-4\varepsilon)\Delta} \cdot 2^{2\Delta} \leq e^{-\Delta/2}.$$

Hence, with probability as least $1 - e^{-\sqrt{k}/2}$, the number of misses is greater than $2\Delta$. $\qquad\square$

For every $k$ and $\Delta \geq \sqrt{k}$, consider the instance we constructed. By Lemma E.3, the algorithm chooses more than $k + \Delta$ centers to cover all but $\Delta$ light clusters with probability at least $1 - e^{-\sqrt{k}/2}$. Thus, at the time when the algorithm chose $k + \Delta$ centers, the number of uncovered light clusters was greater than $\Delta$. Hence, in the clustering with $k + \Delta$ centers sampled by $k$-means++, the total cost is at least $\frac{1}{4} \ln(k/\Delta)$, while the cost of the optimal solution with $k$ clusters is 1. For every $k$ and $\Delta < \sqrt{k}$, the total cost is at least $\frac{1}{4} \ln(k/\Delta')$ with $\Delta' = \sqrt{k}$ extra centers, which concludes the proof. $\qquad\square$

# F   Exponential Race $k$-means++ and Reservoir Sampling

In this section, we show how to implement $k$-means++ algorithm in parallel using $R$ passes over the data set. This implementation, which we refer to as $k$-means++$_{\text{ER}}$ (exponential race $k$-means++), is very similar to $k$-means$\|$, but has stronger theoretical guarantees. Like $k$-means$\|$, in every round, $k$-means++$_{\text{ER}}$ tentatively selects $\ell$ centers, in expectation. However, in the same round, it removes some of the just selected centers (without making another pass over the data set). Consequently, by the end of each iteration, the algorithm keeps at most $k$ centers.

We can run $k$-means++$_{\text{ER}}$ till it samples exactly $k$ centers; in which case, the distribution of $k$ sampled centers is identical to the distribution of the regular $k$-means++, and the expected number of rounds or passes over the data set $R$ is upper bounded by

$$O\left(\frac{k}{\ell} + \log \frac{\text{OPT}_1(\mathbf{X})}{\text{OPT}_k(\mathbf{X})}\right).$$

We note that $R$ is never greater than $k$. We can also run this algorithm for at most $R^*$ rounds. Then, the expected cost of the clustering is at most

$$5(\ln k + 2)\,\text{OPT}_k(\mathbf{X}) + 5R^*\left(\frac{4k}{e\ell R^*}\right)^{R^*} \cdot \text{OPT}_1(\mathbf{X}).$$

## F.1   Algorithm

In this section, we give a high level description of our $k$-means++$_{\text{ER}}$ algorithm. In Section F.2, we show how to efficiently implement $k$-means++$_{\text{ER}}$ using lazy updates and explain why our algorithm makes $R$ passes over the data set.

The algorithm simulates $n$ continuous-time stochastic processes. Each stochastic process is associated with one of the points in the data set. We denote the process corresponding to $x \in \mathbf{X}$ by $P_t(x)$. Stochastic process $P_t(x)$ is a Poisson process with variable arrival rate $\lambda_t(x)$.

The algorithm chooses the first center $c_1$ uniformly at random in $\mathbf{X}$ and sets the arrival rate of each process $P_t(x)$ to be $\lambda_t(x) = \text{cost}(x, \{c_1\})$. Then, it waits till one of the Poisson processes $P_t(x)$ jumps. When process $P_t(x)$ jumps, the algorithm adds the point $x \in \mathbf{X}$ (corresponding to that process) to the set of centers $C_t$ and updates the arrival rates of all processes to be

$$\lambda_t(y) = \text{cost}(y, C_t)$$

for all $y \in \mathbf{X}$. Note that if $y$ is a center, then the arrival rate $\lambda_t(y)$ is 0.

The algorithm also maintains a round counter $R$. In the lazy version of this algorithm (which we describe in the next section), the algorithm makes a pass over the data set and samples a new batch of centers every time this counter is incremented. Additionally, at the end of each round, the algorithm checks if it chose at least one center in that round, and in the unlikely event that it did not, it selects one center with probability proportional to the costs of the points.

Initially, the algorithm sets $R = 0$, $t_0 = 0$, and $t_1 = \ell/\text{cost}(\mathbf{X}, \{c_1\})$. Then, at each time point $t_i$ ($i \geq 1$), we increment $R$ and compute

$$t_{i+1} = t_i + \ell/\text{cost}(\mathbf{X}, C_{t_i}),$$

where $C_{t_i}$ is the set of all centers selected before time $t_i$. We refer to the time frame $[t_{i-1}, t_i]$ for $i \geq 1$ as the $i$-th round. The algorithm stops when one of the following conditions holds true (1) the number of sampled centers is $k$; or (2) the round counter $R$ equals the prespecified threshold $R^*$, which may be finite or infinite.

Before analyzing this algorithm, we mention that every Poisson process $P_t$ with a variable arrival rate $\lambda_t$ can be coupled with a Poisson process $Q_s$ with rate 1. To this end, we substitute the variable

$$s(t) = \int_0^t \lambda_\tau d\tau,$$

and let

$$P_t \equiv Q_{s(t)}.$$

Observe that the expected number of arrivals for process $Q_s$ in the infinitesimal interval $[s, s + ds]$ is $ds = \lambda_t dt$ which is exactly the same as for process $P_t$.

It is convenient to think about the variables $s$ as "current position", $t$ as "current time", and $\lambda_t$ as "current speed" of $s$. To generate process $P_t(x)$, we can first generate Poisson process $Q_s(x)$ with arrival rate 1 and then move the position $s_t(x)$ with speed $\lambda_t(x)$. The process $P_t(x) = Q_{s_t(x)}(x)$ is a Poisson process with variable arrival rate $\lambda_t(x)$.

**Theorem F.1.** *I. If the number of rounds is not bounded (i.e., $R^* = \infty$), then the distribution of centers returned by k-means++$_{ER}$ is identical to the distribution of centers returned by k-means++.*

*II. Moreover, the expected number of rounds $R$ is upper bounded by*

$$(1 + o_k(1)) \cdot \left( \lceil \frac{k}{\ell} \rceil + \log \frac{2 \operatorname{OPT}_1(\mathbf{X})}{\operatorname{OPT}_k(\mathbf{X})} \right),$$

*and never exceeds $k$.*

*III. If the threshold $R^*$ is given ($R^* < \infty$), then the cost of the solution after $R^*$ rounds is upper bounded by*

$$5(\ln k + 2) \operatorname{OPT}_k(\mathbf{X}) + 5R^* \left( \frac{4k}{e\ell R^*} \right)^{R^*} \cdot \operatorname{OPT}_1(\mathbf{X}).$$

*Proof of Part I.* For the sake of analysis, we assume that after the algorithm outputs solution $C$, it does not terminate, but instead continues to simulate Poisson processes $P_t(x)$. It also continues to update the set $C_t$ (but, of course, not the solution) and the arrival rates $\lambda_t(x)$ till the set $C_t$ contains $k$ centers. Once $|C_t| = k$, the algorithm stops updating the set of centers $C_t$ and arrival rates but still simulates continuous-time processes $P_t(x)$. Clearly, this additional phase of the algorithm does not affect the solution since it starts after the solution is already returned to the user.

We prove by induction on $i$ that the first $i$ centers $c_1, \ldots, c_i$ have exactly the same joint distribution as in k-means++. Indeed, the first center $c_1$ is drawn uniformly at random from the data set $\mathbf{X}$ as in k-means++. Suppose centers $c_1, \ldots, c_i$ are already selected. Then, we choose the next center $c_{i+1}$ at the time of the next jump of one of the Poisson processes $P_t(x)$. Observe that the conditional probability that a particular process $P_t(x)$ jumps given that one of the processes $P_t(y)$ ($y \in \mathbf{X}$) jumps is proportional to $\lambda_t(x)$, which in turn equals the current $\operatorname{cost}(x, C_t)$ of point $x$. Hence, the distribution of center $c_{i+1}$ is the same as in k-means++. This completes the proof of item I. $\qquad\square$

*Proof of Part II.* We now show items II and III. Define process

$$P_t(\mathbf{X}) = \sum_{x \in \mathbf{X}} P_t(x).$$

Its rate $\lambda_t(\mathbf{X})$ equals $\sum_{x \in \mathbf{X}} \lambda_t(x)$. We couple this process with a Poisson $Q_s(\mathbf{X})$ with arrival rate 1 as discussed above. We want to estimate the number of centers chosen by the algorithm in the first $R'$ rounds. To this end, we count the number of jumps of the Poisson process $P_t(\mathbf{X})$ (recall that we add a new center to $C_t$ whenever $P_t(\mathbf{X})$ jumps unless $|C_t|$ already contains $k$ centers). The number of jumps equals $P_{t_{R'}}$ which, in turn, equals $Q_{s_{R'}}$ where $s_{R'}(\mathbf{X})$ is the position of $s(\mathbf{X})$ at time $t_{R'}$:

$$s_{R'}(\mathbf{X}) = \int_0^{t_{R'}} \lambda_\tau(\mathbf{X}) \, d\tau = \sum_{i=0}^{R'-1} \int_{t_i}^{t_{i+1}} \lambda_\tau(\mathbf{X}) \, d\tau \geq \sum_{i=0}^{R'-1} (t_{i+1} - t_i) \cdot \lambda_{t_{i+1}}(\mathbf{X}).$$

Here, we used that $\lambda_t(\mathbf{X})$ is non-increasing, and thus, $\lambda_{t_{i+1}}(\mathbf{X}) \leq \lambda_\tau(\mathbf{X})$ for all $\tau \in [t_i, t_{i+1}]$. We now recall that $(t_{i+1} - t_i) = \ell/\operatorname{cost}(\mathbf{X}, C_{t_i})$ and $\lambda_{t_{i+1}}(\mathbf{X}) = \operatorname{cost}(\mathbf{X}, C_{t_{i+1}})$. Hence,

$$s_{R'}(\mathbf{X}) \geq \ell \sum_{i=0}^{R'-1} \frac{\operatorname{cost}(\mathbf{X}, C_{t_{i+1}})}{\operatorname{cost}(\mathbf{X}, C_{t_i})}.$$

By the inequality of arithmetic and geometric means, we have

$$s_{R'}(\mathbf{X}) \geq \ell \cdot R' \left( \prod_{i=0}^{R'-1} \frac{\text{cost}(\mathbf{X}, C_{t_{i+1}})}{\text{cost}(\mathbf{X}, C_{t_i})} \right)^{1/R'} = \ell \cdot R' \left( \frac{\text{cost}(\mathbf{X}, C_{t_{R'}})}{\text{cost}(\mathbf{X}, C_{t_0})} \right)^{1/R'} \tag{5}$$

$$= \ell \cdot R' \left( \frac{\text{cost}(\mathbf{X}, C_{t_{R'}})}{\text{cost}(\mathbf{X}, \{c_1\})} \right)^{1/R'}.$$

We now use this equation to prove items II and III. For item II, we let random variable $R'$ to be

$$R' = 2e\lceil k/\ell \rceil + \log \frac{\text{cost}(\mathbf{X}, \{c_1\})}{\text{OPT}_k(\mathbf{X})}.$$

Note that $R'$ depends on the first center $c_1$ (which is chosen in the very beginning of the algorithm) but not on the Poisson processes $P_t(x)$. Since, $C_t$ always contains at most $k$ centers, we have $\text{cost}(x, C_{t_{R'}}) \geq \text{OPT}_k(\mathbf{X})$, and consequently

$$s_{R'}(\mathbf{X}) \geq \ell \cdot R' \left( \frac{\text{OPT}_k(\mathbf{X})}{\text{cost}(\mathbf{X}, \{c_1\})} \right)^{1/R'} > \ell \cdot 2e\lceil k/\ell \rceil \cdot 1/e \geq 2k.$$

The expected number of jumps of the Poisson process $Q_s(\mathbf{X})$ in the interval $[0, s_{R'}(\mathbf{X})]$ equals $Q_{s_R(\mathbf{X})}(\mathbf{X})$. Observe that

$$Q_{s_R(\mathbf{X})}(\mathbf{X}) \geq Q_{2k}(\mathbf{X})$$

and $Q_{2k}(\mathbf{X})$ is a Poisson random variable with parameter $2k$. By the Chernoff bound[3], it makes fewer than $k$ jumps with exponentially small probability in $k$; namely, with probability at most $(e/2)^{-k}$. Thus, with probability at least $1 - (e/2)^{-k}$, the algorithm selects $k$ centers in the first $R'$ rounds. Moreover, if it does not happen in the first $R^*$ rounds, then it selects $k$ centers by the end of the second $R'$ rounds again with probability at least $1 - (e/2)^{-k}$ and so on. Hence, the expected number of rounds till it selects $k$ centers is $(1 + o_k(1))R'$. Finally, observe that the expectation of $\text{cost}(\mathbf{X}, \{c_1\})$ over the choice of the first center equals $2\,\text{OPT}_k(\mathbf{X})$. Since $\log(\cdot)$ is a convex function, we have

$$\mathbb{E}[R'] \leq 2e\lceil k/\ell \rceil + \log \frac{2\,\text{OPT}_1(\mathbf{X})}{\text{OPT}_k(\mathbf{X})}.$$

Therefore, we showed that the expected number of rounds is upper bounded by the right hand side of the expression above times a multiplicative factor of $(1 + o_k(1))$. A slightly more careful analysis gives a bound of

$$(1 + o_k(1)) \left( e\lceil k/\ell \rceil + \log \frac{2\,\text{OPT}_1(\mathbf{X})}{\text{OPT}_k(\mathbf{X})} \right).$$

This concludes the proof of item II. $\qquad\qquad\qquad\qquad\qquad\qquad\qquad\qquad\qquad$ □

*Proof of Part III.* We now prove item III. Denote $T = t_{R^*}$. Consider the event

$$\mathcal{E} = \{\text{algorithm samples } k \text{ centers in the first } R^* \text{ rounds}\}.$$

Let $\bar{\mathcal{E}}$ be the complimentary events to $\mathcal{E}$. Then,

$$\mathbb{E}\big[\text{cost}(\mathbf{X}, C_T)\big] = \mathbb{E}\big[\text{cost}(\mathbf{X}, C_T) \cdot \mathbb{1}(\mathcal{E})\big] + \mathbb{E}\big[\text{cost}(\mathbf{X}, C_T) \cdot \mathbb{1}(\bar{\mathcal{E}})\big].$$

We now separately upper bound each of the terms on the right hand side. It is easy to upper bound the first term:

$$\mathbb{E}[\text{cost}(\mathbf{X}, C_T) \cdot \mathbb{1}(\mathcal{E})] \leq 5(\ln k + 2) \cdot \text{OPT}_k(\mathbf{X}),$$

because the distribution of centers returned by $k$-means++$_{\text{ER}}$ is identical to the distribution of centers returned by $k$-means++. We now bound the second term. Denote by $\mathcal{D}_\rho$ the event

$$\mathcal{D}_\rho = \left\{ \text{cost}(\mathbf{X}, C_T) \geq \left( \frac{\rho k}{\ell R^*} \right)^{R^*} \text{cost}(\mathbf{X}, \{c_1\}) \right\}.$$

We prove the following claim.

**Claim F.2.** *The following inequality holds for every real number $\rho \in [1, \ell R^*/k]$ and any choice of the first center $c_1$:*

$$\Pr\left(\bar{\mathcal{E}} \text{ and } \mathcal{D}_\rho \mid c_1\right) \leq e^{-(\rho-1)k}\rho^k.$$

*Proof.* We use inequality (5) with $R' = R^*$:

$$s_{R^*}(\mathbf{X}) \geq \ell \cdot R^* \left(\frac{\text{cost}(\mathbf{X}, C_T)}{\text{cost}(\mathbf{X}, \{c_1\})}\right)^{1/R^*}.$$

It implies that $s_{R^*}(\mathbf{X}) \geq \rho k$ if event $\mathcal{D}_\rho$ occurs. On the other hand if $\bar{\mathcal{E}}$ occurs, then the number of centers chosen by the end of round $R^*$ is less than $k$ and, consequently, the number of jumps of $P_t(\mathbf{X})$ in the interval $[0, T]$ is less than $k$:

$$P_T(\mathbf{X}) \equiv Q_{s_{R^*}(\mathbf{X})}(\mathbf{X}) < k.$$

Hence, we can bound $\Pr(\bar{\mathcal{E}} \text{ and } \mathcal{D}_\rho \mid c_1)$ as follows:

$$\Pr(\bar{\mathcal{E}} \text{ and } \mathcal{D}_\rho) \leq \Pr\left(\mathcal{D}_\rho \text{ and } Q_{s_{R^*}}(\mathbf{X}) < k \mid c_1\right) \leq$$
$$\leq \Pr\left(\mathcal{D}_\rho \text{ and } Q_{\rho k}(\mathbf{X}) < k \mid c_1\right) \leq \Pr\left(Q_{\rho k}(\mathbf{X}) < k \mid c_1\right).$$

Random variable $Q_{\rho k}(\mathbf{X})$ has the Poisson distribution with parameter $\rho k$ and is independent of $c_1$. By the Chernoff bound, the probability that $Q_{\rho k}(\mathbf{X}) \leq k$ is at most (as in Part II of the proof):

$$\Pr\left\{Q_{\rho k}(\mathbf{X}) \leq k\right\} \leq e^{-\rho k}\left(\frac{e\rho k}{k}\right)^k = e^{-(\rho-1)k}\rho^k.$$

This completes the proof of Claim F.2. $\qquad\qquad\qquad\qquad\qquad\qquad\qquad\qquad\qquad\qquad\square$

Let

$$Z = \left(\frac{\ell R^*}{k}\right)^{R^*} \cdot \frac{\text{cost}(\mathbf{X}, C_T)}{\text{cost}(\mathbf{X}, \{c_1\})}.$$

Then, by Claim F.2,

$$\Pr\{\bar{\mathcal{E}} \text{ and } Z \geq \rho^{R^*} \mid c_1\} \leq e^{-(\rho-1)k}\rho^k. \tag{6}$$

Write,

$$\mathbb{E}\left[\mathbb{1}(\bar{\mathcal{E}}) \cdot Z \mid c_1\right] = \int_0^\infty \Pr\left(\mathbb{1}(\bar{\mathcal{E}}) \text{ and } Z \geq r \mid c_1\right) dr \leq 1 + \int_1^\infty \Pr\left(\mathbb{1}(\bar{\mathcal{E}}) \text{ and } Z \geq r \mid c_1\right) dr.$$

We now substitute $r = \rho^{R^*}$ and then use (6):

$$\mathbb{E}\left[Z \cdot \mathbb{1}(\bar{\mathcal{E}}) \mid c_1\right] \leq 1 + R^* \int_1^\infty \Pr\left(\bar{\mathcal{E}} \text{ and } Z \geq \rho^{R^*} \mid c_1\right) \cdot \rho^{R^*-1} d\rho$$

$$\leq 1 + R^* \int_1^\infty e^{-(\rho-1)k}\rho^{k+R^*-1} d\rho.$$

We note that $R^* \leq k$, since our algorithm chooses at least one center in each round. Thus, by Lemma F.3 (which we prove below), the integral on the right hand side is upper bounded by $eR^*/2 \cdot (4/e)^{R^*}$. Hence,

$$\mathbb{E}\left[Z \cdot \mathbb{1}(\bar{\mathcal{E}}) \mid c_1\right] \leq 1 + \frac{eR^*}{2} \cdot \left(\frac{4}{e}\right)^{R^*}.$$

Multiplying both sides of the inequality by $(k/\ell R^*)^{R^*} \cdot \text{cost}(\mathbf{X}, \{c_1\})$ and taking the expectation over $c_1$, we get the desired inequality:

$$\mathbb{E}\left[\text{cost}(\mathbf{X}, C_T) \cdot \mathbb{1}(\bar{\mathcal{E}})\right] \leq \left(1 + \frac{eR^*}{2}\left(\frac{4}{e}\right)^{R^*}\right)\left(\frac{k}{\ell R^*}\right)^{R^*} \mathbb{E}_{c_1}\left[\text{cost}(\mathbf{X}, \{c_1\}\right]$$

$$= \left(1 + \frac{e}{2}R^*\left(\frac{4}{e}\right)^{R^*}\right)\left(\frac{k}{\ell R^*}\right)^{R^*} \text{OPT}_1(\mathbf{X})$$

$$< 5R^*\left(\frac{4k}{e\ell R^*}\right)^{R^*} \text{OPT}_1(\mathbf{X}).$$

This finishes the proof of Theorem F.1. $\qquad\qquad\qquad\qquad\qquad\qquad\qquad\qquad\qquad\qquad\square$

**Lemma F.3.** *For $R^* < k$, we have*

$$\int_1^\infty e^{-(\rho-1)k} \rho^{k+R^*-1} d\rho \leq \frac{e}{2}\left(\frac{4}{e}\right)^{R^*}.$$

*Proof.* Since $e^{-(\rho-1)}\rho \leq 1$ for all $\rho \geq 1$, we have $e^{-(\rho-1)k}\rho^k \leq e^{-(\rho-1)R^*}\rho^{R^*}$ for any $R^* < k$. Thus, we have

$$\int_1^\infty e^{-(\rho-1)k} \rho^{k+R^*-1} d\rho \leq \int_1^\infty e^{-(\rho-1)R^*} \rho^{2R^*-1} d\rho = e^{R^*} \int_1^\infty e^{-\rho R^*} \rho^{2R^*-1} d\rho$$

$$= e^{R^*} \int_1^\infty (e^{-\rho}\rho^2)^{R^*} \rho^{-1} d\rho.$$

Observe that $e^{-\rho}\rho^2 \leq 4/e^2$ for any $\rho \geq 1$. Hence,

$$(e^{-\rho}\rho^2)^{R^*} = (e^{-\rho}\rho^2)^{R^*-1} \cdot e^{-\rho}\rho^2 \leq (4/e^2)^{R^*-1} e^{-\rho}\rho^2.$$

Thus,

$$\int_1^\infty e^{-(\rho-1)k} \rho^{k+R^*-1} d\rho \leq \frac{4^{R^*-1} \cdot e^{R^*}}{e^{2(R^*-1)}} \cdot \int_1^\infty e^{-\rho}\rho \, d\rho = \frac{4^{R^*-1}}{e^{R^*-2}} \cdot \frac{2}{e} = \frac{e}{2}\left(\frac{4}{e}\right)^{R^*}.$$

$\square$

## F.2 Lazy Implementation of $k$-means++$_{\text{ER}}$

We now describe how we can efficiently implement the $k$-means++$_{\text{ER}}$ algorithm using a lazy reservoir sampling. We remind the reader that the time of the first jump of a Poisson process with parameter $\lambda$ is distributed as the exponential distribution with parameter $\lambda$. Imagine for a moment, that the arrival rates of our Poisson processes were constant. Then, in order to select the first $k$ jumps, we would generate independent exponential random variables with parameters $\lambda(x)$ for all $x$ and choose $k$ smallest values among them. This algorithm is known as the reservoir sampling(see Efraimidis & Spirakis (2006)). To adapt this algorithm to our needs, we need to update the arrival rates of the exponential random variables. Loosely speaking, we do so by generating exponential random variables with rate 1 for Poisson processes $Q_s(x)$ which are described above and then updating the speeds $\lambda_t(x)$ of variables $s_t(x)$. We now formally describe the algorithm.

In the beginning of every round $i$, we recompute costs of all points in the data set. Then, we draw an independent exponential random variable $S_x$ with rate 1 for every point $x$, and let $S_t(x) = S_x$. We set

$$\tau_t(x) = \frac{S_t(x)}{\lambda_t(x)}.$$

Think of $S_t(x)$ as the distance $s_t(x)$ needs to travel till process $Q_s(x)$ jumps; $\lambda_t(x)$ is the speed of point $s_t(x)$; and $\tau_t(x)$ is the time left till $Q_s(x) = P_t(x)$ jumps if the speed $\lambda_t$ does not change. Among all points $x \in X$, we select a tentative set of centers $Z$ for this round. The set $Z$ contains all points $x$ with $t_{i-1} + \tau_t(x) \leq t_i$. This is the set of all points for which their Poisson processes would jump in the current round if their arrival rates remained the same till the end of the round. Since the arrival rates can only decrease in our algorithm, we know for sure that for points $x$ outside of $Z$, the corresponding processes $P_t(x)$ will not jump in this round. Thus, we can safely ignore those points during the current round.

We also note that in the unlikely event that the initial set $Z$ is empty, we choose $x$ with the smallest time $\tau_t(x)$ and add it to the set of centers $C_t$. (This is equivalent to choosing a point with probability proportional to $\text{cost}(x, C_t)$ by the memorylessness property of the exponential distribution).

The steps we described above – updating costs $\text{cost}(x, C_t)$, drawing exponential random variables $S_x$, and selecting points in the set $Z$ – can be performed in parallel using one pass over the data set. In the rest of the current round, our algorithm deals only with the set $Z$ whose size in expectation is at most $\ell$ (see below).

While the set $Z$ is not empty we do the following. We choose $x \in Z$ with the smallest value of $\tau_t(x)$. This $x$ corresponds to the process that jumps first. Then, we perform the following updates: We add

$x$ to the set of centers $C_t$. We set the "current time" $t$ to $t = t' + \tau_{t'}(x)$, where $t'$ is the time of the previous update. If $x$ is the first center selected in the current round, then we let $t'$ to be the time when the round started (i.e., $t_{i-1}$). We recompute the arrival rates (speeds) $\lambda_t(x)$ for each $x$ in $Z$. Finally, we update the values of all $\tau_t(x)$ for $x \in Z$ using the formula

$$\tau_t(x) = \frac{S_t(x) - \lambda_{t'}(x) \cdot (t - t')}{\lambda_t(x)},$$

here $\lambda_{t'}(x) \cdot (t - t')$ is the distance variable $s_t(x)$ moved from the position where it was at time $t'$; $S_t(x) - \lambda_{t'}(x) \cdot (t - t')$ is the remaining distance $s_t(x)$ needs to travel till the process $Q_t(x)$ jumps; and $\tau_t(x)$ is the remaining time till $P_t(x)$ jumps if we do not update its arrival rate. After we update $\tau_t(x)$, we prune the set $Z$. Specifically, we remove from set $Z$ all points $x$ with $t + \tau_t(x) > t_i$. As before, we know for sure that if $x$ is removed from $Z$, then the corresponding processes $P_t(x)$ will not jump in the current round.

This algorithm simulates the process we described in the previous section. The key observation is that Poisson processes $P_t(x)$ we associate with points $x$ removed from $Z$ cannot jump in this round and thus can be safely removed from our consideration. We now show that the expected size of the set $Z$ is at most $\ell$. In the next section, we analyze the running time of this algorithm.

Then we show that the expected size of the set $Z$ in the beginning of each round $i + 1$ is at most $\ell$. Since every point $x$ belongs to $Z$ with probability

$$\Pr\{x \in Z\} = \Pr\left\{\frac{\mathcal{S}_x}{\text{cost}(x, C_{t_i})} \leq \frac{\ell}{\text{cost}(\mathbf{X}, C_{t_i})}\right\} = \Pr\left\{\mathcal{S}_x \leq \ell \cdot \frac{\text{cost}(x, C_{t_i})}{\text{cost}(\mathbf{X}, C_{t_i})}\right\}.$$

The right hand side is the probability that the Poisson process $Q_s(x)$ with rate 1 jumps in the interval of length $\ell \cdot \text{cost}(x, C_{t_i})/\text{cost}(\mathbf{X}, C_{t_i})$ which is upper bounded by the expected number of jumps of $Q_s(x)$ in this interval. The expected number of jumps exactly equals $\ell \cdot \text{cost}(x, C_{t_i})/\text{cost}(\mathbf{X}, C_{t_i})$. Thus, the expected size of $Z$ is upper bounded as

$$\mathbb{E}|Z| = \sum_{z \in \mathbf{X}} \Pr\{z \in Z\} \leq \sum_{z \in \mathbf{X}} \ell \cdot \frac{\text{cost}(z, C_{t_i})}{\text{cost}(\mathbf{X}, C_{t_i})} = \ell.$$

### F.3 Run time

According to our analysis above, the number of new centers chosen at each round of $k$-means++$_{\text{ER}}$ is at most the size of set $Z$, which is $O(\ell)$ with high probability. In the beginning of every round, we need to update costs of all data points, which requires $O(n\ell d)$ time. In each round, we also need to maintain the rates of all points in set $Z$, which needs $O(\ell^2 d)$ time. Thus, the total running time for $k$-means++$_{\text{ER}}$ with $R$ rounds is $O(Rn\ell d)$. We note that before running our algorithm, we can reduce the dimension $d$ of the space to $O(\log k)$ using the Johnson–Lindenstrauss transform (see Johnson & Lindenstrauss (1984)). This will increase the approximation factor by a factor of $(1 + \varepsilon)$ but make the algorithm considerably faster (see Makarychev et al. (2019), Becchetti et al. (2019), and Boutsidis et al. (2010)).