[Reviews · NeurIPS 2020]

Review 1

Summary and Contributions: I should mention that I have reviewed this paper before (submitted at another venue). The contribution is clearly stated in the title. This is theoretical paper that improves the theoretical approximation guarantees of the k-means++ and k-means|| algorithms. Even though the improvements may be small but these improvement are made using some interesting collection of techniques that itself may be interesting for many readers. Here are some comments about the writeup: 1. It may help the readers if you mention that the algorithms being discussed are for the discrete setting (where centers are chosen from among the points) and also work for metric setting. Note that some of the previous work mentioned in the "related work" section are for the Euclidean case. 2. Lines 47-60: Does it make sense to also experiment with k-means|| where $\ell * T = k$? This may also help ruling out explanation (2). 3. You may want to check: https://proceedings.icml.cc/static/paper_files/icml/2020/3965-Paper.pdf In the sense that the quality of the centers may also be measured in terms how how much improvement local search gives. 3. The Martingale based analysis is also found in the previous work such as Aggarwal, Deshpande, and Kannan. You should add this citation in relevant places. I have read the author response.

Strengths: - k-means++ and k-means|| are simple algorithms that are used in practice. Understanding approximation guarantees for these algorithms help us understand why they work so well in practice. The paper gives a nice compilation of results and techniques used for the analysis of these simple algorithms and is well-written. This could be of value to many attendees who are interested in these algorithms.

Weaknesses: I do not see any significant weaknesses in the paper.

Correctness: Yes.

Clarity: Yes. The paper is well-presented and it is not too difficult to go over the high-level structure of the arguments presented.

Relation to Prior Work: Yes

Reproducibility: Yes

Additional Feedback:


Review 2

Summary and Contributions: In this submission new bounds on the approximation factor of k-means++ and k-means|| are presented. k-means++ is a well-known seeding method of Lloyd's method with approximation factor O(log(k)) and k-means|| is by now also a well-known scalable variant of k-means++. The first theoretical contribution is an upper bound of 5(ln(k)+2) on the approximation factor of k-means++, which improved upon the previously known upper bound of 8(ln(k)+2) by a constant factor. Then bicriteria approximations are considered, i.e., one uses k-means++ and k-means|| to compute clusterings with k+Delta clusters for some Delta > 0 and compares the costs of these clusterings with those of an optimal k-clustering. Both for k-means++ and k-means|| such bicriteria results are already known but in this submission improved bounds are shown. These improved bounds exhibit almost the same asymptotic behavior with respect to Delta as the known bounds but the constants are significantly smaller. The improvement of the approximation factor of k-means++ from 8(ln(k)+2) to 5(ln(k)+2) is based on proving a better bound on the expected cost of covered clusters (i.e., those clusters from which at least one center is chosen). The rest of the proof follows more or less along the same lines as various analyses of k-means++ in the literature. I find it interesting that apparently it has not been observed before that the analysis of Arthur and Vassilvitskii can be improved in this respect. In the submission also an example is presented showing that 5*ln(k) is essentially a tight bound. The proofs of the bicriteria results are more involved. They also use some known ingredients from analyses in the literature but improve these analyses in many aspects. In a nutshell both analyses boil down to proving that as long as the uncovered clusters have high costs it is very likely that they will get covered. In the appendix also experiments are shown with k-means++, k-means|| and a variant of k-means++ with pruning (in this variant more than k centers are sampled by k-means++ and then only k of these are selected). k-means|| and k-means++ with pruning yield clusterings with essentially the same quality and perform slightly better than k-means++. The conclusion one can draw from this is that oversampling seems to be very helpful in obtaining good clusterings.

Strengths: I think the line of research is very interesting and up-to-date. It is very relevant to the NeurIPS community in my opinion. The proposed analyses are clever and introduce many new ideas though some arguments are along the lines of known analyses in the literature. The experiments show that oversampling is probably the reason why k-means|| performs better in experiments than k-means++, which is also interesting (but not very surprising). Since k-means++ and k-means|| are well-known and successful algorithms both in theory and in practice, enhancing the understanding of their behavior is very important.

Weaknesses: One might consider as a weakness that nothing very novel is shown. The bounds on the approximation guarantees are improved but in the end only the constants get better and the asymptotic behavior is not changed (due to some lower bounds, one cannot expect better asymptotic bounds). There is also no essentially new analysis technique presented but the known techniques are applied in a more clever way than before.

Correctness: I did not have time to check all details in the appendix but all results seem very plausible and everything I checked is correct.

Clarity: The paper is generally well written and easy to follow.

Relation to Prior Work: Yes, the relevant literature is cited and discussed.

Reproducibility: Yes

Additional Feedback:


Review 3

Summary and Contributions: This is an interesting paper which provides a large list of improved approximation bounds for the k-means++ and k-means|| (k-means parallel) algorithms, and some variants thereof. After providing a thorough introduction into the algorithms and related results, the author(s) state their contributions: (1) an improved bound on the multiplicative factor by which the k-means++ output differs from the optimal k-means clustering solution, (2) improved bounds on an oversampled form of k-means++, and (3) improved bounds on a form of k-means|| without pruning. Inside the main text, the authors provide the crucial parts of proofs of the three results stated above. These results are substantiated in the supplementary material, which is quite extensive. The proof techniques are interesting and enjoyable to read. Some numerics are also presented to provide explanation of why the k-means|| algorithm works better than k-means++.

Strengths: - k-means++ and its variants are standard and useful algorithms, and this paper contains many nice results which advance the current knowledge of the approximation guarantees of these methods. The - The proofs are interesting to read and utilize a fairly broad set of techniques. The writing is clear and consistent throughout.

Weaknesses: - My main concern for the paper is that the organization is a bit awkward. The results in Sections 4-6 each rely on additional results provided in the supplementary material, but the statements of these deferred proofs often appear abruptly (e.g. Lemma 4.2 and Lemma C.1 both appear in the middle of the proof of Lemma 4.1). A few lines/sections appear without motivation and could likely be moved to the appendix (more below in the "Clarity" section). This could make space to have some discussions in the main text that guide the reader through the list of proofs in the supplementary materials. Alternatively, the author(s) could add a short discussion at the end summarizing some of the insights that made this work possible, e.g. in contrast to the proof techniques of (Arthur and Vassilvitskii 2007).

Correctness: I have gone through all the details in the main text and verified that they are correct. The proofs in the appendix appear correct, but I have not verified these details except for a few of the lemmas. The short empirical section seems rigorous, although code is not provided to reproduce these figures.

Clarity: Overall, the paper is very well-written. Here are some nitpicks: - something seems to be off with the latex formatting, as many lines are missing line numbers - p5.L129: why mention Markov chain here? It can be mentioned in the appendix, where it is actually used. Otherwise the reader may expect to see more details about the Markov chain comment in the main text. - Does Section 3.1 belong closer to the current Section 6, where it is actually used? It could also be moved to the appendix and referenced from within Section 6.

Relation to Prior Work: Yes - comparisons of approximation bounds are clearly provided and explained.

Reproducibility: Yes

Additional Feedback: -------- -------- EDIT after author feedback and discussion: Overall this seemed like a nice paper with interesting results, and I am happy to continue supporting it after seeing the author feedback and discussion. I do encourage the authors to think some more about the suggestions re: organization. --------


Review 4

Summary and Contributions: This paper presents tighter analyses for the approximation bounds of the classical k-means++ and k-means|| algorithms. In particular, the original k-means++ analysis yielding an approximation of 8(ln(k) + 2) from the optimal solution is here improved to 5(ln(k) + 2). Similarly, the k-means|| bound is improved by a constant factor. Both k-means++ and k-means|| are ordinarily used by practitioners thanks to their simplicity, so in that respects these improvements are nice to have. On the other hand, the improvements on the analyses are only marginal and not substantive enough in my opinion to pass the bar for acceptance. What would make the paper a lot stronger is a consolidation of the analyses for k-means++ and k-means||. To elaborate, k-means++ runs for k iterations picking a single center per iteration (proportionally to the squared distance to the closest already chosen center); whereas k-means|| can be seen as a natural generalization in that it runs for T iterations and picks \ell centers per iteration. What would be nice is to study k-means|| for general T and \ell, so to obtain a single bound that recovers both the k-means++ performance guarantee (for T=k, \ell=1), as well as the k-means|| bound (for \ell=\Theta(k)). Note that the bound presented for k-means|| is immaterial in the case when T=O(k) and \ell=O(1). Such an analysis would also underline other sweet spots for these parameters (A complementary empirical study of it could also be useful.) Minor: - To help the reader, please add a reference to the NP-hardness of k-means.

Strengths: - Applicability: k-means++ and k-means|| are both widely used. - Claims are sound

Weaknesses: Improvements are marginal and individual to each algorithm.

Correctness: Claims are sound to me. I couldn't spot any obvious issues in the proofs.

Clarity: The paper is well written and results are presented properly.

Relation to Prior Work: It would have been good to expand on the work by Rozhon (2020) since it's directly related to this one.

Reproducibility: Yes

Additional Feedback:

[Author Response · NeurIPS 2020]

We would like to thank all the reviewers for their detailed reviews and valuable comments. We will address all the comments in the revised version of the paper. We will now discuss some common issues raised by the reviewers and then move to specific comments.

Several reviewers mentioned that our paper improves approximation guarantees for $k$-means++ by a constant factor. We want to point out that we improve the bi-criteria approximation ratio for $k$-means++ very substantially in the regime where the number of additional centers is small. This regime is important because it is the one to which practical heuristics for determining $k$ (like the elbow method) might lead to. More specifically, when the number of additional centers is $\Delta = \frac{k}{\log k}$, our approximation guarantee is $O(\log \log k)$ while the $k$-means++ bound by Arthur and Vassilvitskii (2007) and the bi-criteria bounds by Aggarwal, Deshpande, and Kannan (2009) and Wei (2016) give only an $O(\log k)$ approximation. Thus, in this regime of parameters, our paper provides approximation guarantees that are substantially stronger than previously known.

As was pointed out by Reviewer 2, the bounds on the approximation factor for *non-bi-criteria* $k$-means++ due to Arthur and Vassilvitskii (2007) are tight up to a constant factor. Their upper bound is $8 \ln k + 2$, and their lower bound is $2 \ln k$. Since $k$-means++ and $k$-means$\|$ are extensively used in practice, we believe it is really important to narrow down the gap between upper and lower bounds even further. Our paper does so by improving the upper bound from $8 \ln k + 2$ to $5 \ln k + 2$. Moreover, our results (specifically, Lemma 4.1) can be used to get similar improvements for many other papers on $k$-means++ and its variants. We also show that our bound of 5 for Lemma 4.1 is tight.

Finally, let us mention that our paper not only gives better approximation guarantees for $k$-means$\|$ than the paper by Bahmani, Moseley, Vattani, Kumar, and Vassilvitskii (2012) but also provides a simpler analysis.

**Reviewer 1:** We ran some experiments with $\ell \cdot T = k$. The performance was similar to $k$-means++. Thank you for pointing us to the "$k$-means++: Few More Steps Yield Constant Approximation" paper. It is a very interesting paper, and we will cite it in the revised version. We currently cite Aggarwal, Deshpande, and Kannan (2009) in the introduction. We will cite this paper in other relevant places as well (including the martingale analysis).

**Reviewer 3:** Thank you for the detailed suggestions. We will reorganize the paper to improve its readability.

**Reviewer 4:** Thank you for the detailed comments. We agree that a more unified analysis for $k$-means++ and $k$-means$\|$ would be nice to have. But one bottleneck for achieving this is that although the marginal distributions for picking individual points are the same for each round in $k$-means++ and $k$-means$\|$ when $\ell = 1$ and $T = k$, the joint distributions are quite different. In each round, $k$-means++ picks *exactly* one center whereas $k$-means$\|$ can pick any number of centers.

Regarding the paper by Rozhon (2020): We got to know about this work only after the list of accepted papers for ICML 2020 came out (which was very close to the NeurIPS deadline). So we did not get much time to compare our work with that paper. We will certainly do this in the revised version of our paper. In general, the guarantees proved in that paper for $k$-means$\|$ are orthogonal to our guarantees.

We will add references to the NP-hardness and APX-hardness of $k$-means results.

[Meta-Review · NeurIPS 2020]

Although the improved analysis does not offer a huge jump compared to the known bounds, the importance of this algorithm and the simplification of the proof makes the paper an important contribution. Please give some discussion on how your results are orthogonal to the recent results by Rozhon (we agree that they are orthogonal).